# Repeated ketamine anesthesia during neurodevelopment upregulates hippocampal activity and enhances drug reward in male mice

Jianchen Cui [1,2,3,4,5,11], Xianshu Ju[1,2,3,11], Yulim Lee[1,2,3,4], Boohwi Hong[6,7], Hyojin Kang[8], Kihoon Han [9], Won-Ho Shin[10], Jiho Park[6], Min Joung Lee[1,2,3], Yoon Hee Kim[6,7], Youngkwon Ko[6,7], Jun Young Heo [1,2,4✉] & Woosuk Chung [1,4,6,7✉]

Early exposures to anesthetics can cause long-lasting changes in excitatory/inhibitory synaptic transmission (E/I imbalance), an important mechanism for neurodevelopmental disorders. Since E/I imbalance is also involved with addiction, we further investigated possible changes in addiction-related behaviors after multiple ketamine anesthesia in late postnatal mice. Postnatal day (PND) 16 mice received multiple ketamine anesthesia (35 mg kg$^{-1}$, 5 days), and behavioral changes were evaluated at PND28 and PND56. Although mice exposed to early anesthesia displayed normal behavioral sensitization, we found significant increases in conditioned place preference to both low-dose ketamine (20 mg kg$^{-1}$) and nicotine (0.5 mg kg$^{-1}$). By performing transcriptome analysis and whole-cell recordings in the hippocampus, a brain region involved with CPP, we also discovered enhanced neuronal excitability and E/I imbalance in CA1 pyramidal neurons. Interestingly, these changes were not found in female mice. Our results suggest that repeated ketamine anesthesia during neurodevelopment may influence drug reward behavior later in life.

[1] Department of Medical Science, Chungnam National University School of Medicine, Daejeon, South Korea. [2] Department of Biochemistry, Chungnam National University School of Medicine, Daejeon, South Korea. [3] Infection Control Convergence Research Center, Chungnam National University School of Medicine, Daejeon, South Korea. [4] Brain Korea 21 FOUR Project for Medical Science, Chungnam National University, Daejeon, South Korea. [5] Department of Anesthesiology, The First People's Hospital of Yunnan Province. The Affiliated Hospital of Kunming University of Science and Technology, Kunming, China. [6] Department of Anesthesiology and Pain Medicine, Chungnam National University Hospital, Daejeon, South Korea. [7] Department of Anesthesiology and Pain Medicine, Chungnam National University School of Medicine, Daejeon, South Korea. [8] Division of National Supercomputing, Korea Institute of Science and Technology Information (KISTI), Daejeon, South Korea. [9] Department of Neuroscience, College of Medicine, Korea University, Seoul, South Korea. [10] Cell Model Research Group, Department of Predictive Toxicology, Korea Institute of Toxicology, Daejeon, South Korea. [11] These authors contributed equally: Jianchen Cui, Xianshu Ju. ✉email: junyoung3@gmail.com; woosuk119@cnu.ac.kr

As millions of children receive anesthesia for surgical or diagnostic procedures every year, anesthesia-induced neurotoxicity in the developing brain has received considerable research interest in recent decades[1]. Preclinical studies have shown that anesthetics commonly used in clinical settings may act as neurotoxins during neurodevelopment, inducing widespread neuronal cell death or excitatory/inhibitory (E/I) imbalance[2–5]. The disruption of E/I balance during the critical neurodevelopmental period is of considerable concern since E/I imbalance is an important mechanism for neurodevelopmental disorders[6–8]. In line with such concerns, studies that reported anesthesia-induced E/I imbalance also found long-term behavioral changes in activity, anxiety, sociability, learning, and memory[5,9–12].

Based on preclinical studies, clinical studies have attempted to identify long-term negative effects of anesthesia in young children. While recent prospective clinical studies suggested that a short anesthetic exposure does not affect general intelligence[13–15], the same studies also reported increased behavioral problems based on parental reports[16]. Other studies have reported possible associations between early anesthetic exposures and neurodevelopmental disorders (autism, ADHD)[17,18]. However, it is important to note that anesthesia-induced neurotoxicity was first identified in animal models without direct clinical evidence, making it difficult to speculate how this might manifest in children. Recent studies have acknowledged this fact and performed a wide range of tests to evaluate diverse cognitive and behavioral aspects. While these studies discovered changes in specific behaviors (motor and social linguistic performance, emotions) and executive functions[16,19], many behavioral aspects remain unstudied. One possible, but less evaluated, the disorder is addiction. Previous studies have shown that changes in E/I synaptic transmission are deeply involved with addiction[20,21]. Single cocaine exposure has been shown capable of affecting AMPA/NMDA receptor currents[22], and morphine exposure and withdrawal have been shown to affect GABA signaling[23]. Thus, it is possible that the synaptic changes due to early anesthesia exposures may also affect addiction behavior.

Ketamine, an intravenous anesthetic agent, is often used for sedation or anesthesia during short procedures in pediatric patients due to its' lack of respiratory depression and excellent analgesic effects[24]. However, unlike other anesthetic agents, ketamine is also classified as a 'dissociative anesthetic'. By disconnecting the thalamo-neocortical and limbic systems, ketamine can induce hallucinations and altered sensory states. As a result of these attributes, ketamine has come to be used as a recreational drug at sub-anesthetic doses, leading to drug abuse[25]. Since ketamine exposures during neurodevelopment induce E/I imbalance[26,27], and ketamine itself has addictive properties, we hypothesized that repeated ketamine anesthesia during neurodevelopment might affect addiction behavior later in life.

To test our hypothesis, late postnatal mice (postnatal day 16 [PND16]) received multiple anesthetic treatments with ketamine (35 mg kg$^{-1}$, i.p.) for 5 consecutive days. Long-term changes in addiction-related behaviors were evaluated by measuring behavioral sensitization and conditioned place preference (CPP) to low-dose ketamine (20 mg kg$^{-1}$, i.p.) and nicotine (0.5 mg kg$^{-1}$ s.c.) in adolescent (PND28) and adult (8 weeks old) mice. Although there was no change in behavioral sensitization, we discovered increases in place preference for both ketamine and nicotine in male mice who had received early ketamine anesthesia. Interestingly, such behavioral changes did not occur in female mice. To further identify possible mechanisms underlying the changes in drug reward but not sensitization, we performed transcriptome analysis and whole-cell recordings in the hippocampus, a brain region deeply involved with CPP[28–30]. Our results suggest that exposures to ketamine during neurodevelopment can cause long-lasting changes in drug reward, and that sex must be regarded

as an important biological variable when studying the early effects of anesthesia.

## Results

**Development of acute tolerance after repeated ketamine anesthesia in late postnatal male mice.** As previous studies suggest that multiple anesthetic exposures may affect neurodevelopment[15,16,19], we injected late postnatal mice with an anesthetic dose of ketamine (35 mg kg$^{-1}$, i.p.) for 5 consecutive days (PND16–20). The anesthetic dose of ketamine was based on our previous study, where we found that higher doses occasionally caused prolonged anesthesia or mortality (>40 mg kg$^{-1}$, i.p)[26]. Unlike other anesthetic agents, clinical studies have reported acute tolerance after repeated ketamine anesthesia in children[31,32]. Thus, by measuring the loss-of-righting (LOR) reflex, we first examined whether acute tolerance also developed after repeated ketamine anesthesia in young mice. Although ketamine induced a short duration of anesthesia on day 1 (230 ± 203.4 seconds), the duration of anesthesia significantly declined with following injections (Fig. 1a). In fact, due to the development of acute tolerance, LOR did not develop in most mice after day 3 (Fig. 1a). While our results clearly show rapid development of ketamine tolerance in late postnatal mice, there was no effect on general growth, as ketamine-injected mice showed similar weight gain compared with saline-injected mice (Fig. 1b).

**Behavioral sensitization to low-dose ketamine was not affected by early repeated ketamine anesthesia in male mice.** To investigate whether the acutely developed tolerance can influence addiction-related behaviors later in life, we first examined behavioral sensitization to low-dose ketamine 1 week after anesthesia treatment (Fig. 1c, d). Previous studies have shown that repeated injections of low-dose ketamine (20 mg kg$^{-1}$, i.p.) induce behavioral sensitization[33,34], a progressively enhanced behavioral response following repetitive administration of abusive drugs. Behavioral sensitization has been suggested to reflect neuronal adaptations following substance abuse[35]. After confirming that the baseline activity was not affected by multiple ketamine anesthesia (Control-Saline group vs. Anesthesia-Saline group, Fig. 1e), mice were repeatedly injected with ketamine in a 4-day interval (Fig. 1d). Behavioral sensitization occurred in both groups, as total activity significantly increased after five identical ketamine injections (Fig. 1f). We also found that the level of sensitization was comparable between the two groups (Fig. 1f, g). Thus, although repeated ketamine anesthesia-induced rapid tolerance in late postnatal mice, such acute changes did not affect behavioral sensitization to low-dose ketamine.

**Place preference for low-dose ketamine and nicotine is enhanced 1 week after repeated ketamine anesthesia in male mice.** Another commonly used behavioral test for assessing the motivational effects of addictive drugs is the conditioned place preference (CPP) test, which evaluates contextual associations in reward-related behavior[36]. Thus, we next investigated whether early anesthesia influenced place preference for low-dose ketamine (20 mg kg$^{-1}$, i.p.) (Fig. 2a). Place preference developed regardless of anesthesia history, as evidenced by significant increases in the duration spent in the white (ketamine-paired) chamber after conditioning (Fig. 2b). However, mice in the Anesthesia group developed more robust place preference compared with mice in the Control group (Fig. 2c, Supplementary Fig. 1).

To further investigate whether early ketamine anesthesia can also affect the motivational effects of different drugs of abuse, we next repeated the CPP test using nicotine (0.5 mg kg$^{-1}$, s.c.) (Fig. 2d, e). Interestingly, nicotine-induced place preference occurred only in mice in the Anesthesia group (Fig. 2d). There

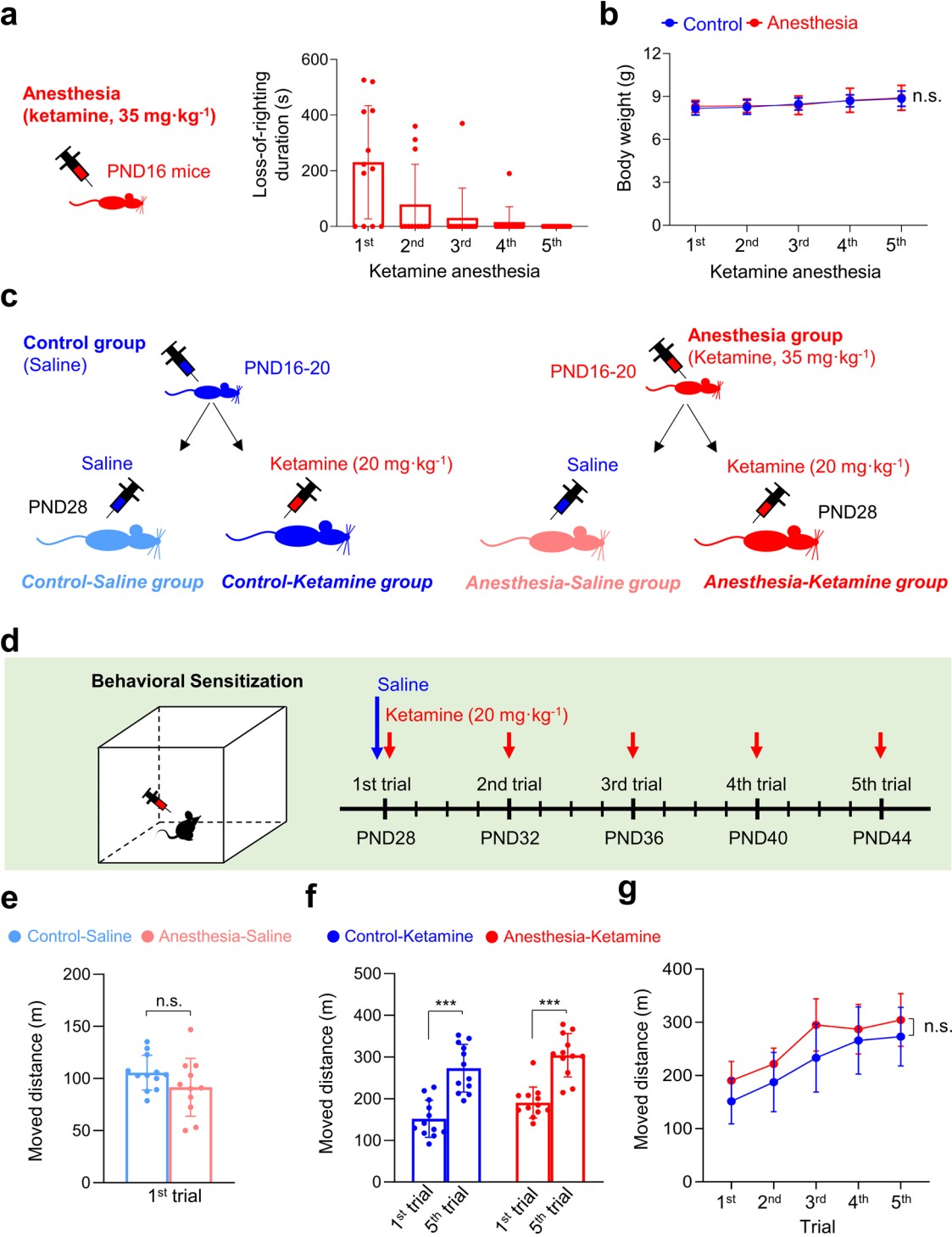

**Fig. 1 Behavioral sensitization to low-dose ketamine is not affected in male mice that received early ketamine anesthesia. a** PND16 male mice were injected with an anesthetic dose of ketamine for 5 consecutive days. The duration of ketamine anesthesia, measured as the LOR reflex duration, was significantly reduced by repeated injections (acute tolerance) ($p = 0.001$, repeated measures [RM]-ANOVA; $n = 12$). **b** Normal weight gain in mice that received repetitive ketamine anesthesia ($p = 0.115$, RM-ANOVA; Control, $n = 15$; Anesthesia, $n = 12$). **c–g** Early repeated ketamine anesthesia does not affect the development of behavioral sensitization to low-dose ketamine. **c, d** Schematic diagram of the experimental design. A behavioral sensitization test was performed 1 week after anesthesia injections. **e** Early ketamine anesthesia did not affect baseline activity at PND28 ($p = 0.158$, Student's $t$ test; Control-Saline, $n = 12$; Anesthesia Saline, $n = 11$). **f** Behavioral sensitization to repeated low-dose ketamine developed in both groups (one-way ANOVA with Tukey multiple comparisons; Control-Ketamine, $n = 12$; anesthesia ketamine, $n = 12$). **g** The development of behavioral sensitization was comparable between the two groups ($p = 0.583$, RM-ANOVA; Control-Ketamine, $n = 12$; Anesthesia ketamine, $n = 12$). Values are presented as means ± SD (n.s., not significant; ***$p < 0.001$).

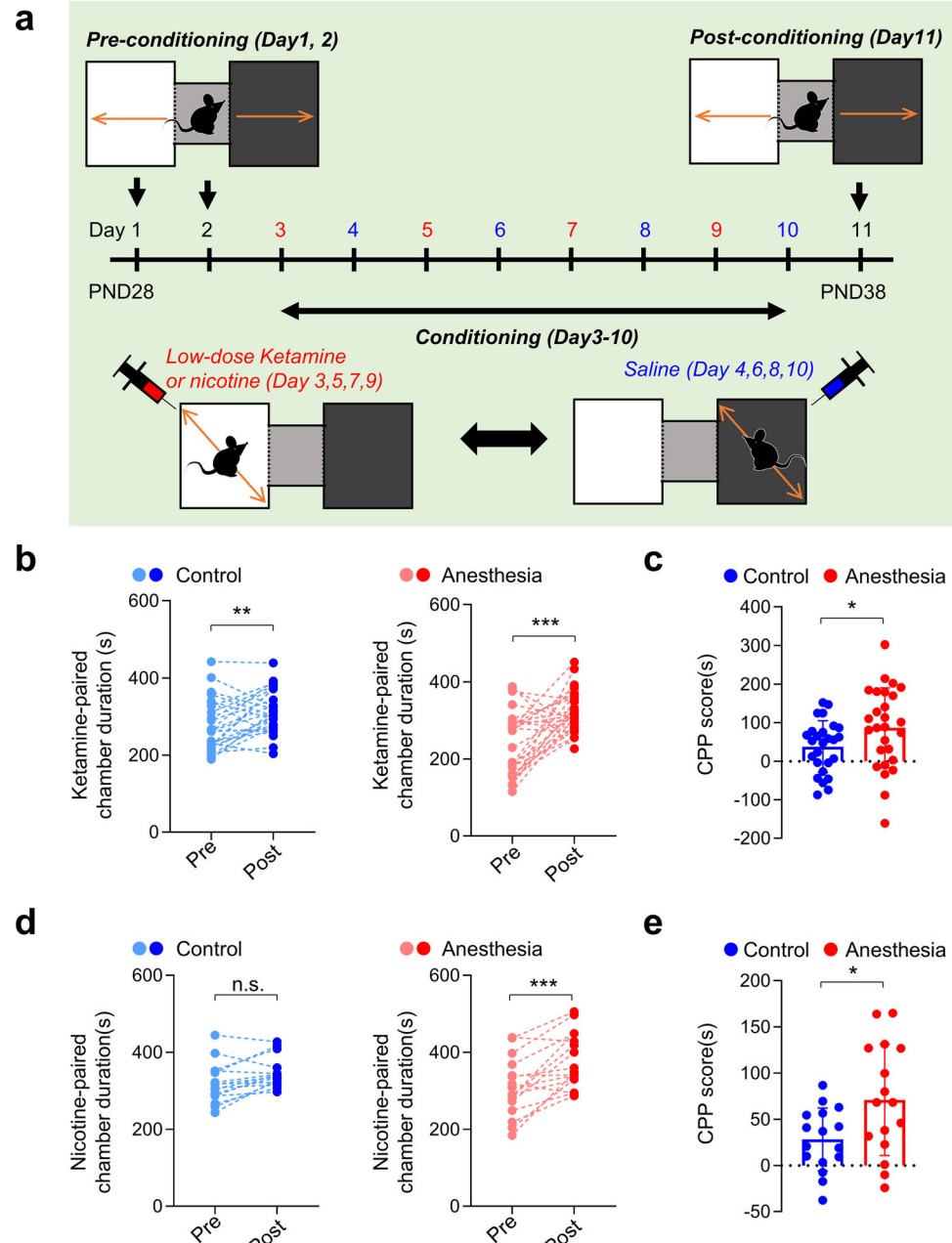

**Fig. 2 Conditioned place preference for low-dose ketamine and nicotine are increased in male mice that received early ketamine anesthesia. a–c** Early repeated ketamine anesthesia increased low-dose ketamine-induced place preference 1 week after anesthesia injections (ketamine CPP). **a** Experimental scheme of ketamine CPP. **b** Time spent in the ketamine-paired (white) chamber was significantly increased after conditioning in both groups (Control, $p = 0.007$; paired $t$ test; $n = 26$ mice; Anesthesia, $p < 0.001$; paired $t$ test; $n = 27$ mice). **c** Summary graph comparing ketamine CPP scores between groups ($p = 0.0496$, Welch ANOVA; control, $n = 26$ mice; Anesthesia, $n = 27$ mice). **d, e** Early repeated ketamine anesthesia increased nicotine-induced place preference 1 week after anesthesia injections (nicotine-CPP). **d** Time spent in the ketamine-paired (white) chamber was significantly increased after conditioning in the Anesthesia group ($p < 0.001$) but not in the Control group ($p = 0.056$) (paired $t$ test; control, $n = 16$ mice; Anesthesia, $n = 16$ mice). **e** Summary graph comparing nicotine-CPP scores between groups ($p = 0.0192$, Welch ANOVA; control, $n = 16$ mice; Anesthesia, $n = 16$ mice.). Values are presented as means ± SD (n.s., not significant; *$p < 0.05$, **$p < 0.05$, ***$p < 0.001$).

was also a significant difference in CPP scores between the two groups (Fig. 2e). These results suggest that early repeated ketamine anesthesia selectively enhances place preference to abusive drugs without affecting behavioral sensitization.

**Place preference for nicotine continues to be enhanced after a prolonged drug-free period in male mice.** We next investigated whether increases in place preference persist after a more

prolonged drug-free period by performing the CPP test 5 weeks after anesthesia treatment (8 weeks old) (Fig. 3a). Unexpectedly, the same CPP protocol with low-dose ketamine did not induce place preference in either group at this age (Fig. 3b, c). As this may be attributable to the lower susceptibility to substance abuse at older ages[37], we repeated the CPP test with a longer conditioning period (14 days). However, place preference still did not develop (Supplementary Fig. 2). Another possibility is that ketamine-induced CPP is not as robust as

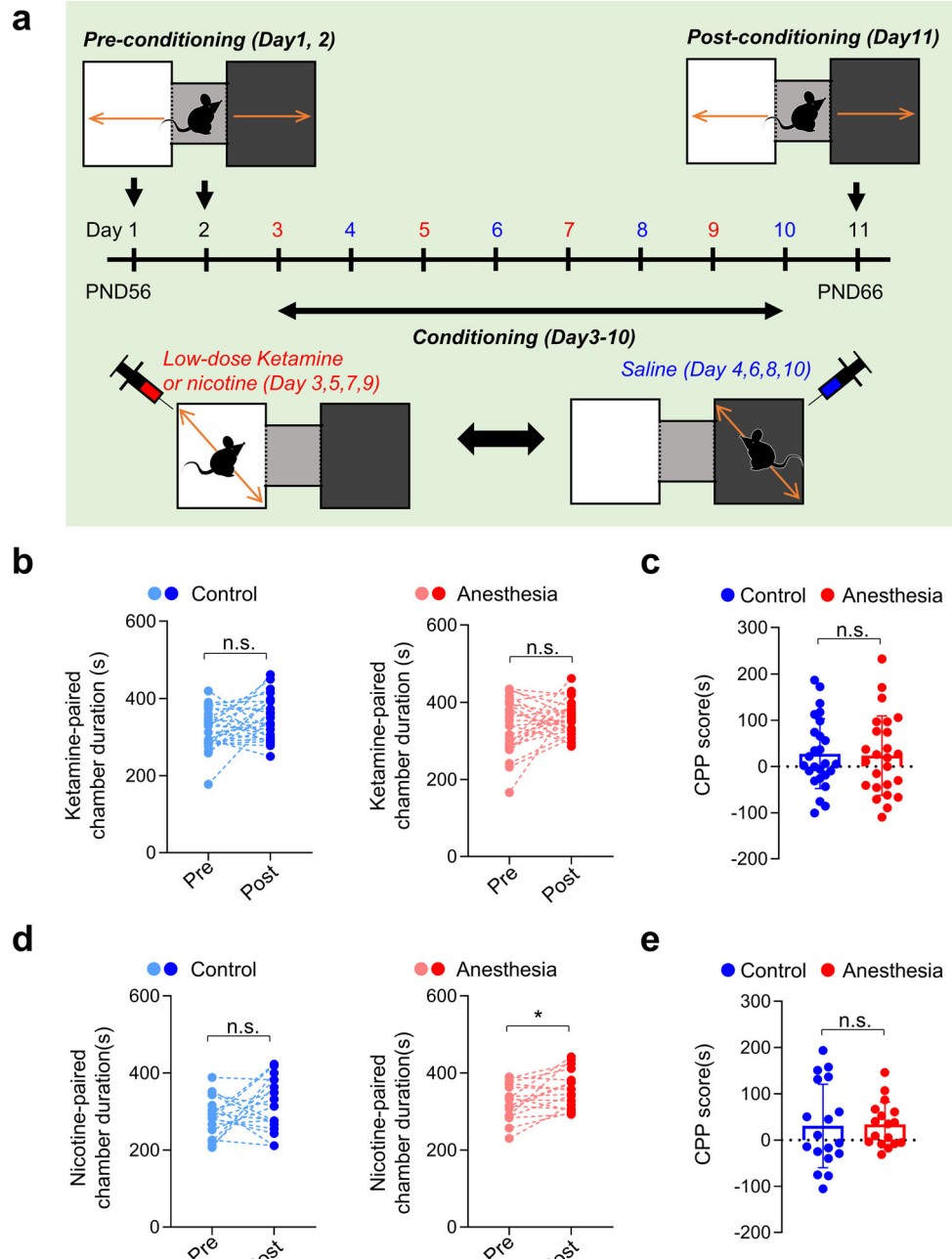

**Fig. 3 Place preference for nicotine is persistently enhanced in adult male mice after early repeated ketamine anesthesia. a–c** Early repeated ketamine anesthesia does not affect place preference for low-dose ketamine in adult mice (ketamine CPP). **a** Experimental scheme of ketamine CPP performed in adult male mice. **b** Place preference to low-dose ketamine does not occur in both Control ($p = 0.074$) and anesthesia ($p = 0.183$) groups (paired $t$ test; control, $n = 26$; anesthesia, $n = 25$). **c** Summary graph comparing ketamine CPP scores between groups ($p = 0.858$, student's $t$ test; control, $n = 26$; anesthesia, $n = 25$). **d–e** Nicotine-induced place preference occurred only in mice that received early repeated ketamine anesthesia (nicotine-CPP). **d** Time spent in the nicotine-paired (white) chamber was significantly increased after conditioning in the anesthesia ($p = 0.011$) but not in the control ($p = 0.172$) group (paired $t$ test; control, $n = 18$, anesthesia, $n = 17$). **e** Summary graph comparing nicotine-CPP scores between groups ($p = 0.894$, Welch ANOVA; control, $n = 18$, Anesthesia, $n = 17$). Values are presented as means ± SD (n.s., not significant; *$p < 0.05$).

other addictive drugs, since a previous study has also reported negative results[38]. The simplicity of our CPP apparatus may also be involved. While most CPP tests use various visual and tactile cues to distinguish chambers, we used a more simplified version (black and white chambers). Although such simplicity may suppress the development of place preference, we considered it could increase the sensitivity for discovering differences in place preference induction. Such simplicity may

also contribute to the lack of place preference for nicotine in the Control group (Figs. 2d, 3d). Importantly, place preference for nicotine still developed in mice that received early ketamine anesthesia (Anesthesia group) (Fig. 3d). Although the CPP score for nicotine was comparable between control and anesthesia mice (Fig. 3e), our results suggest that the effects of early repeated ketamine anesthesia may persist into adulthood.

**Repeated ketamine anesthesia increases the expression of genes encoding axon-localized proteins that regulate neuronal excitability in the hippocampus of male mice.** Given that the mesolimbic dopaminergic system is deeply involved in addiction, we investigated whether early ketamine anesthesia increased the level of dopamine and its metabolite, 3,4-dihydroxyphenylacticacid (DOPAC), in the mesolimbic circuit using high-performance liquid chromatography (HPLC). These analyses showed that early repeated ketamine did not increase the levels of dopamine or DOPAC, measured directly after the CPP test (Supplementary Fig. 3). In fact, the level of dopamine was decreased in the ventral tegmental area (VTA). However, it is important to note that early ketamine anesthesia affected CPP, but not behavioral sensitization. The CPP test measures reward-related learning and memory of drug-context associations[36,39], a process with deeply involves the hippocampus[28–30]. Thus, we performed a transcriptome analysis (RNA-sequencing) of the hippocampus to identify possible mechanisms underlying the increase in place preference. A total of 38 differentially expressed genes (DEGs; 34 upregulated and 4 downregulated) were identified in mice that received repeated ketamine anesthesia compared with mice in the control group (Fig. 4a, Supplementary Tables 1–3, Supplementary Data 1). However, the majority of changes were less than twofold, suggesting that early ketamine anesthesia does not induce major changes in expression at the single-gene level. To evaluate whether these DEGs are involved in similar biological pathways, we next performed functional enrichment analysis using Enrichr (GO/KEGG pathway analysis). Despite the relatively low number of DEGs, we discovered significant associations with the cellular component categories, "axon initial segment", "main axon", and "node of Ranvier" (Fig. 4b, Supplementary Tables 4–6). There were no changes in molecular function or biological process categories (Fig. 4b, Supplementary Table 5). To further identify changes in biological functions that are driven by large numbers of genes with moderate, but coordinated, changes in expression, we also performed a gene set enrichment analysis (GSEA) using the entire list of genes. Similar to the results of the Enrichr functional enrichment analysis, we found upregulation of genes involved in the regulation of neuronal activity whose encoded products are localized to neuronal axons (Supplementary Tables 7–10). GSEA results also suggested that upregulation occurred mostly in pyramidal neurons of the hippocampus (Supplementary Table 11).

**Repeated ketamine anesthesia affects intrinsic excitability, E/I balance, and neuronal activity in CA1 hippocampal neurons of male mice.** Based on our transcriptome analysis (RNA-sequencing) results, we next evaluated whether repeated ketamine anesthesia affects the electrophysiological properties of CA1 hippocampal pyramidal neurons. Changes in intrinsic neuronal excitability were first evaluated by injecting a series of depolarizing currents under current-clamp conditions[40]. Although there were no differences in resting membrane potential, input resistance, or action potential firing thresholds between groups (Supplementary Fig. 4), we discovered a significant increase in the number of action potential firings (current-firing curve) in the Anesthesia group (Fig. 4c). We also found significant changes in spontaneous synaptic transmission. Repeated ketamine anesthesia increased the frequency of sEPSCs without affecting their amplitude (Fig. 4d), while decreasing the frequency of sIPSCs (Fig. 4e). However, there were no differences in the amplitude of both sEPSCs and sIPSCs (Fig. 4d, e). Interestingly, such changes in synaptic transmission were activity-dependent, as the addition of tetrodotoxin (TTX), a voltage-gated sodium channel blocker, eliminated the changes (mEPSCs/mIPSCs) (Supplementary Fig. 5a–d). Since our results suggest an increase

in neuronal activity, we next evaluated changes in spontaneous neuronal firing under current-clamp ($I = 0$ mode). Although rarely discovered in the control group, spontaneous action potentials were significantly increased in mice that received repeated ketamine anesthesia (Fig. 4f). Our RNA-seq results and electrophysiology data suggest that early repeated ketamine anesthesia induces long-lasting changes in hippocampal activity, which may act as an important mechanism for the increases in place preference behavior.

**Early repeated ketamine anesthesia does not enhance ketamine-induced CPP and hippocampal excitability in female mice.** Sex is now recognized as an important biological variable in neuroscience, and sex-dependent changes in cocaine addiction behavior were recently reported after prolonged antidepressant ketamine treatment in young mice[41]. Thus, we further investigated the effects of early repeated ketamine anesthesia in late postnatal female mice. Similar to male mice, acute tolerance developed during repeated ketamine anesthesia (Supplementary Fig. 6a), and there was no difference in behavioral sensitization to low-dose ketamine 1 week after anesthesia (Supplementary Fig. 6b). However, we found differences in the CPP test results between male and female mice. Although place preference for low-dose ketamine occurred in both groups 1 week after injections (Fig. 5a), in contrast to male mice, place preference for low-dose ketamine was decreased in the Anesthesia group (Fig. 5b). Also, female mice did not develop place preference to nicotine, regardless of anesthesia history (Supplementary Fig. 7). While our results suggest that early ketamine anesthesia does not enhance drug reward behavior in female mice, it is important to recognize that CPP measures drug reward by evaluating the association of a specific context and drug stimulus. Thus, to imply that female mice are resistant to the increase in drug reward induced by early ketamine anesthesia, it is necessary to confirm that non-drug-associated memory is also not affected. By performing the Barnes maze and fear chamber test, we further confirmed that non-drug-associated learning and memory were not affected in female mice (Supplementary Fig. 8).

In line with the sex differences in drug reward behavior, we also found that repeated ketamine anesthesia during neurodevelopment did not affect the electrophysiologic properties of hippocampal neurons in female mice (Fig. 5c–f). We found no changes in intrinsic excitability (Fig. 5c, Supplementary Fig. 9), synaptic transmission (Fig. 5d, e, Supplementary Fig. 10), and spontaneous neuronal activity (Fig. 5f). Although we did not directly investigate the causality between electrophysiological changes and place preference behavior, such sex-differences support our hypothesis that the changes in the hippocampus are involved in the increase in place preference after early ketamine anesthesia.

## Discussion

Recent clinical studies have suggested that exposures to anesthetics during neurodevelopment may affect specific behaviors rather than affecting global cognitive function. In the present study, we investigated possible changes in addiction-related behaviors after repeated ketamine anesthesia in late postnatal mice. We found that repeated ketamine anesthesia did not affect behavioral sensitization, but increased place preference for low-dose ketamine and nicotine. Interestingly, such changes in drug-associated reward behavior only occurred in male mice. In fact, place preference for low-dose ketamine was reduced in female mice that received early ketamine exposures. Consistent with the increase in place preference in male mice, changes in intrinsic excitability, spontaneous synaptic transmission, and a number of

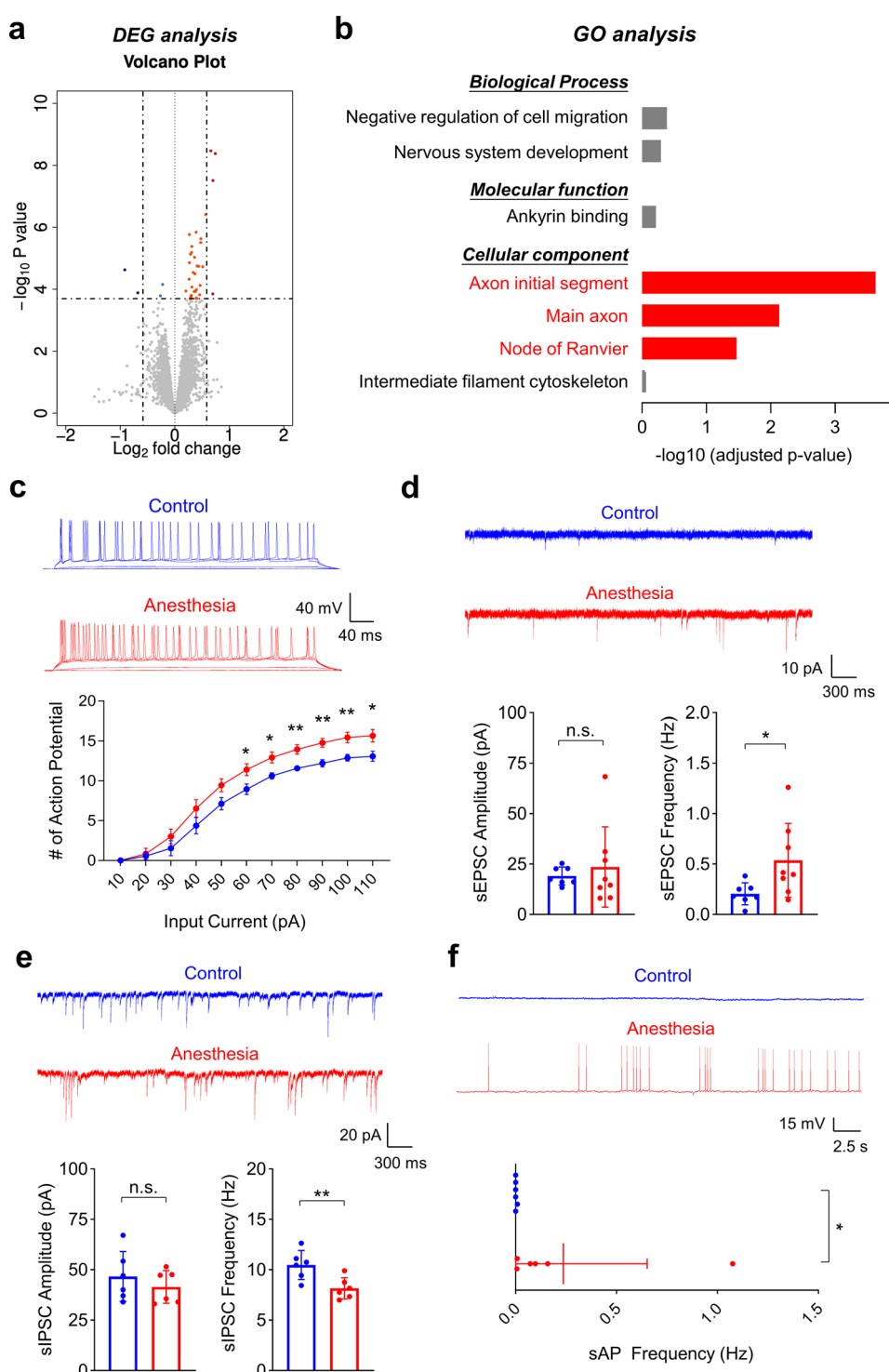

spontaneous action potentials in hippocampal pyramidal neurons also occurred only in male mice.

While few studies have previously investigated the effects of early ketamine exposures on reward behavior[41–43], the present study differs in several important aspects. The first important difference is the dose and duration of ketamine treatment. Previous studies evaluated the consequences of antidepressant doses of ketamine (10–20 mg kg$^{-1}$) which was injected for a prolonged period (10–15 days), whereas we used a higher anesthetic dose of ketamine (35 mg kg$^{-1}$) for a much shorter period (5 days). Secondly, we evaluated changes in both behavioral sensitization and

place preference, unlike previous studies which only measured a single behavior. The third and most important difference is that we identified a plausible mechanism for the male-dependent increase in place preference by discovering male-dependent changes in hippocampal neuronal activity. Recent studies suggested that the dorsal hippocampus plays an essential role in the acquisition of contextual memory associated with abusive drugs such as nicotine and cocaine[28–30]. Our results are in line with such previous studies, as spontaneous neuronal activity was increased in the hippocampus after repeated ketamine anesthesia in male mice.

**Fig. 4 Repeated ketamine anesthesia increases mRNA levels of genes encoding axon-localized proteins, causing changes in hippocampal synaptic transmission and neuronal activity. a, b** RNA-sequencing was performed with hippocampal samples 1 week after repeated ketamine anesthesia or saline injections in male mice. **a** Volcano plot of DEGs, shown as blue (decreased expression) and red (increased expression) circles. **b** Gene ontology (GO) analysis of DEGs. Significant terms are highlighted in red (Axon initial segment, $p < 0.001$; Main axon, $p = 0.007$; Node of Ranvier, $p = 0.034$; $n = 4$ for each group). **c–f** Whole-cell recordings were performed 1 week after repeated ketamine anesthesia in hippocampal CA1 pyramidal neurons of male mice. **c** Enhanced intrinsic excitability after early repeated ketamine anesthesia. The number of action potentials was measured after injecting a series of depolarizing currents under current-clamp conditions (current injection $= 60$ pA, $p = 0.034$; 70 pA, $p = 0.018$; 80 pA, $p = 0.006$; 90 pA, $p = 0.005$; 100 pA, $p = 0.009$; 110 pA, $p = 0.029$; Student's $t$ test; control, $n = 6$ mice [total 20 cells]; anesthesia, $n = 7$ mice [total 20 cells]). **d, e** Altered excitatory and inhibitory spontaneous synaptic transmissions in hippocampal CA1 pyramidal neurons. **d** Anesthesia group showed increased frequency (Welch ANOVA, $p = 0.039$), but normal amplitude (Kruskal–Wallis test, $p = 0.908$) of sEPSCs (control, $n = 7$ mice [total 21 cells]; anesthesia, $n = 8$ mice [total 21 cells]). **e** Anesthesia group showed decreased frequency (Student's $t$ test, $p = 0.010$) but normal amplitude (Student's $t$ test, $p = 0.418$) of sIPSCs in hippocampal CA1 pyramidal neurons (control, $n = 6$ mice [total 22 cells]; anesthesia, $n = 6$ mice [total 22 cells]). **f** Spontaneous action potentials (sAP) are significantly increased after early repeated ketamine anesthesia. sAPs were measured under $I = 0$ mode ($p = 0.010$, Kruskal–Wallis test; Control, $n = 6$ mice [total 21 cells]; anesthesia, $n = 6$ mice [total 24 cells]). Values are presented as means ± SD (n.s., not significant; *$p < 0.05$; **$p < 0.01$).

Although sex is recognized as an important biological factor[44], we did not initially perform experiments in female mice. This was based on our recent study, where we reported non-sex-dependent changes after repeated ketamine anesthesia (two-way ANOVA)[26]. However, another recent study reported conflicting results by showing that repeated ketamine treatment in young mice enhanced place preference for cocaine only in male mice[41]. Interestingly, we also discovered increased drug reward behavior and enhanced hippocampal activity only in male mice. Our results, however, are in contrast with other preclinical studies reporting enhanced sensitivity to ketamine in female rodents[45–48]. For instance, studies have reported that the dose-response threshold for ketamine's antidepressant effect was lower in female mice[45–47]. Another study reported that behavioral sensitization developed only in female mice when using ketamine 2.5 mg kg$^{-1}$ [48]. However, studies using a relatively higher dose of ketamine (10 mg kg$^{-1}$) have reported longer antidepressant effects in male mice[46], and that prolonged injections induced antidepressant effects only in male mice[49]. Based on these results, it is possible that the sex differences in ketamine sensitivity are dose-dependent. Unfortunately, further studies are necessary as only a few studies have focused on higher doses of ketamine (20 mg kg$^{-1}$)[38].

Another interesting finding was the increased expression of genes encoding axon-localized proteins involved with neuronal activity after early ketamine exposures. Early studies have reported increased dendritic spinogenesis after exposure to anesthetics, including ketamine, in late postnatal mice[27]. However, our transcriptome analysis of the hippocampus after ketamine anesthesia revealed expression changes only in genes encoding axon-localized proteins. One possible explanation for such differences is the timing of the experiments. Because the increase in dendritic spines after ketamine exposure is transient[27], such changes may not be detected by RNA-sequencing 1 week after anesthetic exposure. The number of ketamine anesthetic exposures may also be involved, given that previous studies evaluated the effects of a single exposure[27]. While additional studies at different time points using diverse anesthesia protocols are necessary, our results suggest that future studies may need to focus on long-lasting axonal changes rather than transient increases in dendritic spines.

The developmental stage of the animal is an important factor for anesthesia-induced neurotoxicity[2–4,12]. Although the majority of preclinical studies have been performed at PND7, it has been suggested that this age is comparable to a third-trimester human fetus from a neurodevelopment standpoint (www.translatingtime.org)[50,51]. This may cause discrepancies between preclinical and clinical studies since the majority of clinical studies focused on infants or very young children. Our study performed ketamine anesthesia at PND16–20, an age that has been suggested to be comparable to a 6 to 9 months-old human infant[50,51]. However, because it is difficult to directly compare neurodevelopment between humans and rodents, additional studies at diverse ages are necessary to understand the potential effects of early anesthesia.

There are several important limitations in the present study. The first limitation is the lack of evidence confirming a direct causative relationship between the increased hippocampal excitability and changes in reward behavior. Although this can be achieved using direct drug treatment through surgically implanted cannulas[52] or optogenetic/chemogenetic inhibition[53,54], we were unable to perform such studies due to a lack of expertise and equipment. Secondly, we cannot exclude the role of other brain regions. Although we focused on the hippocampus since early ketamine exposures only affected place preference (not behavioral sensitization), hippocampal activity is also involved with dopamine release from VTA dopaminergic neurons[55]. Thus, additional studies in diverse brain regions involved with an addictive behavior may provide additional evidence regarding the changes from early ketamine exposures. Thirdly, the interval between early ketamine anesthesia and the CPP tests may not have been sufficient to evaluate long-term changes in drug reward. CPP tests were performed at PND28 and PND56. However, these ages may be comparable to a 16-month infant and a 2-year-old child from a neurodevelopmental standpoint[50,51]. Thus, although considered adolescent and adult mice, these ages may not be sufficient to evaluate the long-term effects of early ketamine anesthesia. Fourthly, the use of a fixed ketamine dose (35 mg kg$^{-1}$) for repetitive anesthesia despite the development of acute tolerance may also act as a limitation. As the purpose of the study was to evaluate the long-term consequences of early ketamine anesthesia, it may have been reasonable to increase the dose of ketamine when LOR did not occur. However, this could complicate data analysis, since each mouse would have received different doses of ketamine. Lastly, although we discovered changes in place preference after repeated ketamine anesthesia, the CPP test is not a direct animal model for addiction behavior. The CPP test is an animal model of drug reward, and the drug is passively administered[39]. In contrast, during the self-administration test, the subject must learn a task that produces acute effects from self-drug administration. Thus, the self-administration paradigm is considered a more direct measurement of drug reinforcement and addiction[39]. Future studies using the self-administration paradigm are necessary to address the effects of early repeated ketamine anesthesia on addictive behavior.

In conclusion, by evaluating late postnatal mice exposed to repeated ketamine anesthesia, we provide evidence that early anesthetic exposures using ketamine increased hippocampal activity, resulting in long-term change in reward-related learning and memory only in male mice. Our results raise the possibility

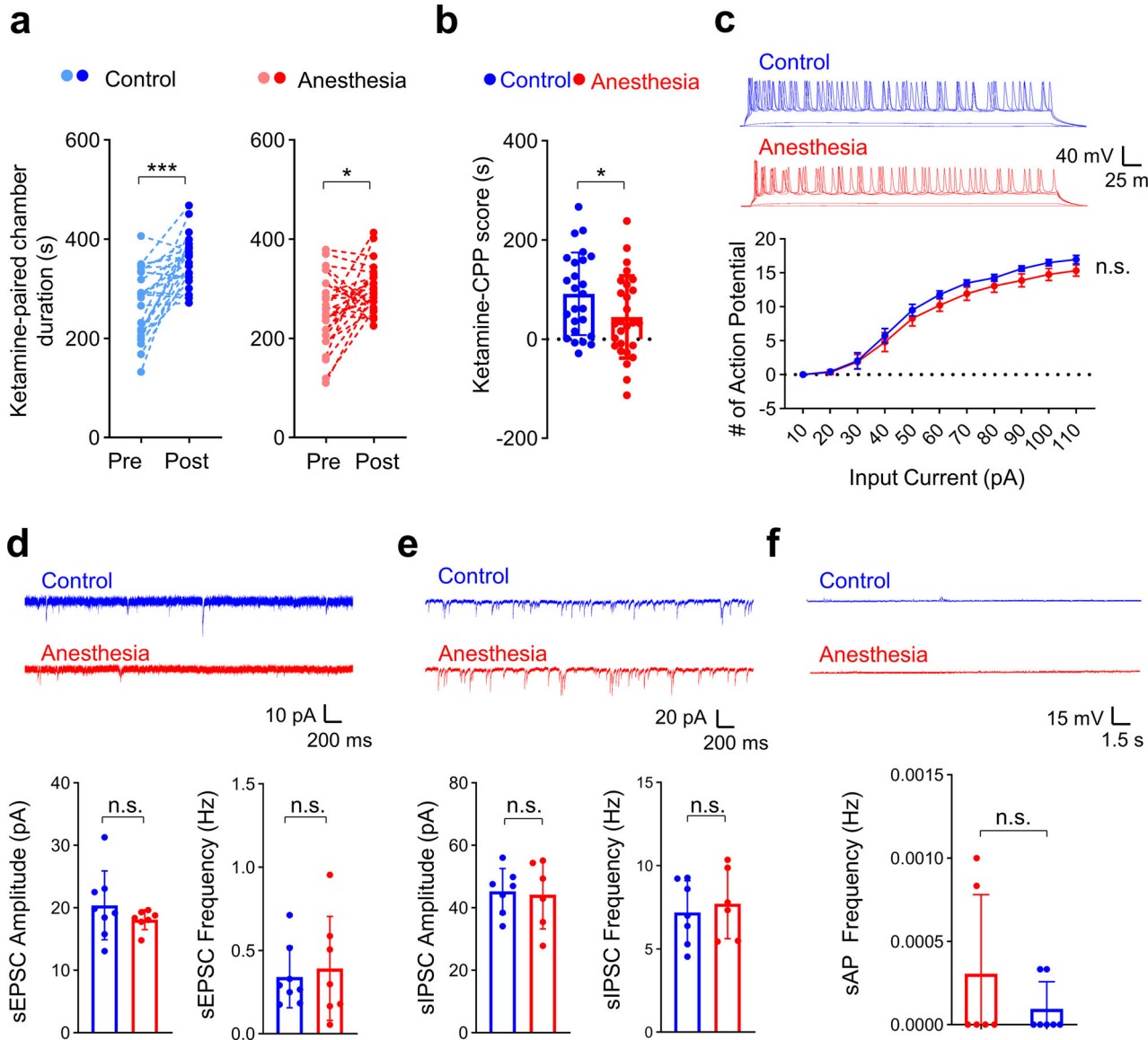

**Fig. 5 Female mice are resistant against the increases in addiction behavior and neuronal changes induced by early repeated ketamine anesthesia.**
**a, b** Early repeated ketamine anesthesia reduces low-dose ketamine-induced place preference 1 week after anesthesia injections in female mice (ketamine CPP). **a** Time spent in the ketamine-paired (white) chamber was significantly increased after conditioning in both groups (control, $p < 0.001$; paired $t$ test; $n = 25$; Anesthesia, $p = 0.010$; paired $t$ test; $n = 27$). **b** Summary graph comparing ketamine CPP scores between groups ($p = 0.047$, Student's $t$ test). **c–f** Whole-cell recordings were performed in hippocampal CA1 pyramidal neurons from female mice 1 week after repeated ketamine anesthesia. **c** Normal intrinsic excitability after early repeated ketamine anesthesia in female mice. The number of action potentials was measured after injecting a series of depolarizing currents under current-clamp conditions (Kruskal–Wallis test and Student's $t$ test; control, $n = 6$ mice (total 20 cells); anesthesia $n = 6$ mice (total 21 cells)). **d, e** Repeated ketamine anesthesia does not affect spontaneous excitatory synaptic transmission (Welch ANOVA, amplitude: $p = 0.190$; Student's $t$ test, frequency: $p = 0.698$; control, $n = 7$ mice (total 21 cells); anesthesia, $n = 8$ mice (total 21 cells)) and spontaneous inhibitory synaptic transmission in female mice (Student's $t$ test, amplitude: $p = 0.833$; Student's $t$ test, frequency: $p = 0.643$; Control, $n = 6$ mice (total 22 cells); anesthesia $n = 7$ mice [total 21 cells]). **f** Early repeated ketamine anesthesia does not affect spontaneous action potentials (sAPs). sAPs were measured under $I = 0$ mode ($p = 0.600$, Kruskal–Wallis test; control, $n = 6$ mice [total 22 cells]; anesthesia, $n = 7$ mice [total 20 cells]). Values are presented as means ± SD (*$p < 0.05$; ***$p < 0.001$; n.s., not significant).

that repeated use of ketamine for anesthetic purposes may affect abusive drug use later in life. Future studies must recognize that early anesthetic exposures may affect diverse behaviors, and that sex should always be considered an important biological variable.

## Methods
**Animals.** All procedures were approved by the Committees on Animal Research at Chungnam National University Hospital (Daejeon, South Korea, CNUH-018-P0006, 202009A-CNU-155). C57BL/6 mice (RRID: MGI:5659186, Damul Science, Daejeon, Korea) were housed at a relative temperature of 22 °C and humidity of

40%, in a 12-hour light/dark cycle, and allowed free access to food and water. Mice were weaned at postnatal day (PND) 21 and housed in groups of three to five per cage.

**Anesthesia.** Ketamine hydrochloride (Huons, South Korea, 50 mg ml$^{-1}$) was diluted to 2.5 mg ml$^{-1}$ with normal saline before injections. Mice were randomly divided into two groups at PND16 (Control and Anesthesia group). Mice in the Anesthesia group received daily intraperitoneal injections of ketamine (35 mg kg$^{-1}$) for 5 consecutive days[26]. Although the majority of preclinical studies exposed young animals to 3 episodes of anesthesia, ketamine is often used outside the operating room for more frequently repeated short-term procedures or radiographic examinations[31,32]. Thus,

we increased the number of daily injections to 5 days. Mice were placed in an empty cage containing fresh bedding (36 °C) for 30 minutes after injections. Mice in the Control group received identical treatment with normal saline.

Anesthesia duration, defined as the time from intraperitoneal injection to the regaining of the righting reflex, was measured using a subset of mice. Once the mice lost the ability to right themselves, mice were placed in a supine position on a flat surface. Righting reflex was defined as the ability to right themselves on all four paws three times within 30 seconds[56].

**Behavior tests**. Behavioral tests (behavioral sensitization, conditioned place preference test, Barnes maze, fear chamber test) were performed in a sound-attenuated room and video-recorded by a blind experimenter. Video recordings were later automatically analyzed using tracking software (Ethovision XT; Noldus Information Technology, Netherlands). Mice with growth disturbance due to malocclusion or severe injury due to fighting in group-housed mice were excluded.

**Behavioral sensitization**. Behavioral sensitization (enhanced locomotor activity after repeated exposures to drugs of abuse) was measured using the open field box (width/height/length, 40 cm). Mice were placed in the center of the open field box after i.p. injection of low-dose ketamine (20 mg kg$^{-1}$) or an equal volume of saline. The general activity was measured for 60 minutes. The experiment was repeated 5 times, in a 4-day interval[57]. Behavioral sensitization was evaluated by measuring the increase in locomotor activity.

**Conditioned place preference (CPP) test**. The CPP apparatus consisted of three chambers connected with small openings (7 × 7 cm). The two main compartments (width/height/length, 15/20/20 cm) differed in color (black vs white) and were connected by a small gray chamber (width/height/length, 10/20/10 cm). The CPP test was performed using a standard CPP protocol including a pre-conditioning phase, a conditioning phase, and a post-conditioning phase. During the pre-conditioning phase (day 1–2), mice were placed in the gray chamber and were allowed to explore the full region of the CPP apparatus freely for 15 minutes. The sessions were video-recorded and the time spent in the white chamber was measured on day 2 (pre-conditioning duration). The conditioning phase was conducted for 8 consecutive days, with one conditioning trial per day. For ketamine CPP, mice received low-dose ketamine (20 mg kg$^{-1}$) and were confined in the white chamber for 30 minutes (days 3, 5, 7, 9). Between low-dose ketamine injections, mice were injected with an equal volume of saline and confined in the black chamber for 30 minutes (days 4, 6, 8, 10). The post-conditioning phase was performed the following day (day 11). Mice were allowed to freely explore the full region of the CPP apparatus freely for 15 minutes, and the time spent in the white (ketamine-paired) chamber was measured (post-conditioning duration). CPP score was calculated using the time spent in the white (ketamine-paired) chamber during pre/post-conditioning phases (CPP score = 'post-conditioning duration'—'pre-conditioning duration'). The same protocol was used for the nicotine-CPP test. Nicotine (nicotine hydrogen tartrate salt, Sigma-Aldrich, St Louis, USA) was subcutaneously injected (0.5 mg kg$^{-1}$).

**Barnes maze test**. Spatial learning and memory were evaluated by performing the Barnes maze test[58]. The maze consisted of a circular platform (92 cm diameter) with 20 holes (5 cm diameter) containing a hidden escape box (Scitech Korea Inc., Korea). The platform was elevated 95 cm above the floor and was surrounded by three spatial cues. The test was performed in a bright room and a buzzer (80 dB) was used to motivate mice to the enter escape box.

Mice were first placed in the escape box for 3 minutes (habituation). Following habituation, mice were placed in the middle of the maze using a black-colored 'start chamber'. 10 seconds after the buzzer was turned on, the start chamber was removed and mice were trained to enter the escape box. The buzzer was turned off once the mice entered the escape box, and the mice remained in the box for 2 minutes (pre-training trial). 1 hour after pre-training, mice received the first training session. Training sessions (three sessions daily) were performed for 4 days. During the trial sessions, the buzzer was turned off once the mice entered the escape box and the mice remained in the escape box for 1 minute. Mice were gently guided if they did not enter the escape box within 3 minutes. To avoid the possibility of intra-maze cues, the maze was rotated each day. After the training sessions, the escape box was removed and mice were placed in the maze for 45 seconds (probe trial).

Learning and memory were compared by measuring the latency, path length, and a number of errors for mice to enter the escape box. Total latency was defined as the duration until mice entered the escape box; total path length was the total moved distance before entering the escape box; total errors were the number of noses pokes over holes irrelevant to the escape box before mice entered the escape hole. As mice may not enter the escape box despite finding the correct escape hole, we also measured the latency, path length, and a number of errors until the mice first encountered the escape hole (primary latency, primary path length, primary errors). Latency, errors, and path length were manually measured by an experimenter blinded to the condition of the mice, while distance was measured using tracking software (Ethovision XT).

**Fear chamber test**. Contextual-fear memory was measured using the fear chamber test (Coulbourn Instruments, MA, USA)[12]. Fear conditioning was performed by applying three trials of a stimulus (1 mA electric shock, 1 minute interval), 5 minutes after placing the mice in the fear chamber. Contextual-fear memory was analyzed the following day after placing mice in the same chamber for 5 minutes. The freezing duration was automatically measured using the FreezeFrame software (Coulbourn Instruments).

**High-performance liquid chromatography (HPLC)**. The level of dopamine and 3,4-dihydroxyphenylacticacid (DOPAC) in several brain regions were measured by high-performance liquid chromatography (HPLC, 1260 Infinity system, Agilent Technologies, CA, USA) with electrochemical detection. After brief anesthesia with sevoflurane, the nucleus accumbens (NAcc), ventral tegmental area (VTA), and striatum were dissected in ice-cold PBS and stored at −80 °C prior to analysis. Brain tissues were homogenized in 0.1 M perchloric acid and centrifuged at 14,000 rpm for 15 minutes. The supernatants were injected using an autosampler and eluted through a SunFire C18 column (4.6 × 100 mm × 5 μm; Waters Corporation, MA, USA). Peaks were detected using an ESA Coulochem III electrochemical detector and analyzed using ChemStation software (Agilent Technologies, CA, USA). The concentrations of dopamine and DOPAC were normalized to total protein per sample determined by a protein assay kit (Thermo Scientific, MA, USA).

**RNA-sequencing**. RNA was extracted from the hippocampus of mice 1 week after repeated saline injections or ketamine anesthesia (4 mice for each group). Total RNA was obtained using the RNeasy Lipid Tissue Mini Kit (QIAGEN, Germany, Cat. 74804) and stored at −80 °C. Library preparation, cluster generation, and sequencing were performed by Macrogen Inc. (Seoul, Korea): Library kit, TruSeq Stranded Total RNA LT Sample Prep Kit (Gold); Library protocol, TruSeq Stranded Total RNA Sample Prep Guide, Part #15031048 Rev. E; Reagent, NovaSeq 6000 S4 Reagent Kit; Sequencing protocol, NovaSeq 6000 System User Guide Document #1000000019358 v02, Sequencer type, NovaSeq 6000; Sequencing Control Software, 1000000019358 v02.

**RNA-Seq analysis**. Transcript abundance was estimated with Salmon (v1.1.0)[59] in Quasi-mapping-based mode onto the Mus musculus genome (GRCm38) with GC bias correction (--gcBias). Quantified gene-level abundance data was imported to R (v.3.5.3) with the tximport[60] package and differential gene expression analysis was carried out using R/Bioconductor DEseq2 (v1.30.1)[61]. Normalized read counts were computed by dividing the raw read counts by size factors and fitted to a negative binomial distribution. The P values were adjusted for multiple testing with the Benjamini–Hochberg correction. Genes with an adjusted P value of less than 0.05 were considered as differentially expressed. The Gene Ontology (GO) enrichment analyses were performed using Enrichr software[62,63]. Mouse gene names were converted to human homologs using the Mouse Genome Informatics (MGI) database (http://www.informatics.jax.org/homology.shtml). Gene Set Enrichment Analysis (GSEA) (http://software.broadinstitute.org/gsea)[64] was used to determine whether a priori-defined gene set would show statistically significant differences in expression between control- and ketamine anesthesia treated mice. Enrichment analysis was performed using GSEAPreranked (gsea-3.0.jar) module on gene set collections downloaded from Molecular Signature Database (MSigDB) v7.0 (http://software.broadinstitute.org/gsea/msigdb). GSEAPreranked was applied using the list of all genes expressed, ranked by the fold change, and multiplied by the inverse of the $p$ value with recommended default settings (1000 permutations and a classic scoring scheme). The False Discovery Rate (FDR) was estimated to control the false-positive finding of a given Normalized Enrichment Score (NES) by comparing the tails of the observed and null distributions derived from 1000 gene set permutations. The gene sets with an FDR of less than 0.05 were considered significantly enriched. The RNA-seq data were submitted to the GEO (Gene Expression Omnibus) repository under accession number GSE175894.

**Electrophysiology (slice preparation)**. Mice were briefly anesthetized with 3% sevoflurane before decapitation. Acute sagittal hippocampal slices (300 μm) were obtained using a VT1200S vibratome (Leica, Switzerland) in ice-cold dissection buffer (212 mM sucrose [Sigma-Aldrich, Cat# S5016], 25 mM NaHCO$_3$ [Sigma-Aldrich, Cat# S6297], 5 mM KCl [Sigma-Aldrich, Cat# P3911], 1.25 mM NaH$_2$PO$_4$ [Sigma-Aldrich, Cat# S0751], 10 mM D-glucose [Sigma-Aldrich, Cat# G7578], 2 mM sodium pyruvate [Sigma-Aldrich, Cat# P2256], 1.2 mM sodium ascorbate [Sigma-Aldrich, Cat# A4034], 3.5 mM MgCl$_2$ [Sigma-Aldrich, Cat# M0250], 0.5 mM CaCl$_2$ [Sigma-Aldrich, Cat# C3881], continuously aerated with 95% O$_2$/5% CO$_2$). The slices were transferred to a chamber filled with warmed (32 °C) artificial cerebrospinal fluid (aCSF: 125 mM NaCl [Sigma-Aldrich, Cat# S7653], 25 mM NaHCO$_3$, 2.5 mM KCl, 1.25 mM NaH$_2$PO$_4$, 10 mM D-glucose, 1.3 mM MgCl$_2$, 2.5 mM CaCl$_2$, aerated with 95% O$_2$/5% CO$_2$) for recovery (30 minutes), and then placed in room temperature prior to the experiments.

**Electrophysiology (whole-cell recordings)**. Whole-cell recordings were obtained in CA1 hippocampal pyramidal neurons[12]. In order to avoid confusion, electrophysiology (whole-cell recordings) studies were performed by a non-blinded

experimenter. However, recordings were blindly analyzed. Brain slices were transferred to a recording chamber and continuously perfused with aerated aCSF maintained at 25–26 °C. Recordings were made using a MultiClamp 700B amplifier (Molecular Devices, USA) and Digidata 1440 A (Molecular Devices) under visual control (BX50WI, Olympus, Japan). Data were acquired with Clampex 11.0.3 (Molecular Devices) and analyzed using Clampfit 11.0.3 (Molecular Devices) blind to the condition of the mice. Ra (access resistance), Rm (membrane resistance), and Cm (membrane capacitance) were measured using the membrane test protocol included in the Clampex 11.0.3 software. Whole-cell recordings were included only in the following conditions: initial Ra <20 MΩ; Rm >100 MΩ; Cm >100 pF; and <20% change in Ra at the end of the recording.

Intrinsic excitability of hippocampal CA1 cells was recorded using a K-gluconate-based internal solution: 137 mM K-gluconate (Sigma-Aldrich, Cat# P1847), 5 mM KCl (Sigma-Aldrich, Cat# P3911), 10 mM HEPES (Sigma-Aldrich, Cat# H3375), 0.2 mM EGTA (Sigma-Aldrich, Cat# E3889), 10 mM Na-phosphocreatine, 4 mM Mg-adenosine triphosphate (ATP, Sigma-Aldrich, Cat# A9187), and 0.5 mM Na-guanosine triphosphate (GTP, Sigma-Aldrich, Cat# G8877), corrected to pH 7.2, 280 mOsm. To inhibit postsynaptic responses, 100 μM Picrotoxin (Sigma-Aldrich, Cat# P1675), 10 μM NBQX (TOCRIS, Cat# 0373), and 50 μM D-AP5 (Alomone Labs, Cat# D-145) were added to the aCSF. Resting membrane potential (RMP) was measured under '$I = 0$ mode' for 1 minute after whole-cell configuration. After stabilization, RMP was adjusted to −70 mV and current inputs were increased from 0 to 110 pA (600 ms, 10 pA increase per sweep, 10-s interval). Spontaneous action potentials (sAP) were measured by perfusing standard aCSF, using the same internal solution. Spontaneous firings were detected under '$I = 0$ mode' for 10 minutes.

The internal solution for recordings of spontaneous excitatory postsynaptic currents (sEPSC) consisted of 117 mM CsMeSO4 (Sigma-Aldrich, Cat# C1426), 10 mM tetraethylammonium chloride (TEA-Cl, Sigma-Aldrich, Cat# T2265), 8 mM NaCl (Sigma-Aldrich, Cat# S9888), 10 mM HEPES (Sigma-Aldrich, Cat# H3375), 5 mM QX314-Cl (TOCRIS, Cat# 2313), 4 mM Mg-adenosine triphosphate (ATP, Sigma-Aldrich, Cat# A9187), 0.3 mM Na-guanosine triphosphate (GTP, Sigma-Aldrich, Cat# G8877), 10 mM EGTA (Sigma-Aldrich, Cat# E3889). For spontaneous inhibitory postsynaptic current (sIPSC) recordings, CsMeSO4 was replaced with 115 mM CsCl (Sigma-Aldrich, Cat# C4036). Neurons were voltage-clamped at −70 mV during the recordings. Picrotoxin (100 μM, Sigma-Aldrich, Cat# P1675) was added for sEPSC recordings, while NBQX (10 μM, TOCRIS, Cat# 0373) and D-AP5 (50 μM, Alomone labs, Cat# D-145) were added for sIPSC recordings. Recording of miniature excitatory postsynaptic currents (mEPSCs) and miniature inhibitory postsynaptic currents (mIPSCs) was separately measured after additionally adding tetrodotoxin (0.5 μM, Abcam, Cat# ab120055).

**Statistics and reproducibility**. Sample sizes were based on the previous studies[12,26,41]. For electrophysiological data, the average of each animal (at least 2 cells per animal) was used. Data were analyzed using the R statistical software package (4.0.3; R Core Team, Austria). All continuous variables were tested for normality and homogeneity of variance. ANOVA (One-way analysis of variance) was performed when both conditions were met. Welch's ANOVA was performed when homogeneity of variance was not met, and the Kruskal–Wallis test was performed if normality was not met. For the CPP test, paired t-test (when normality was met) or Wilcoxon test (when normality was unmet) was used. Repeatedly measured data were such as daily experiments analyzed via Repeated measures analysis of variance (RM-ANOVA) or Friedman rank-sum test. P values < 0.05 were considered significant. Detailed statistic results are described as supplementary data (Supplementary Notes 1–15).

**Reporting summary**. Further information on research design is available in the Nature Research Reporting Summary linked to this article.

## Data availability

Source data underlying figures are presented in Supplementary Data 2. The RNA-seq data are available from the GEO (Gene Expression Omnibus) repository under accession number GSE175894.

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

## Acknowledgements

This work was supported by the National Research Foundation of Korea (NRF), Grant number: NRF- 2017R1A5A2015385, NRF-2018R1C1B6003139, NRF-2019M3E5D1A02068575.

## Author contributions

J.C., X.J. designed, developed, and executed experiments and prepared manuscript; B.H. analyzed the data; Y.L., J.P., W.H.S., H.K. performed experiments; K.H., M.J.L., Y.H.K., Y.K. helped in the design and development of experiments, and in oversight; W.C., J.Y.H. designed, developed, oversaw, and prepared manuscript.

## Competing interests

The authors declare no competing interests.
