## [Peer Review File · Communications Biology]

Reviewers' comments:

Reviewer #1 (Remarks to the Author):

This manuscript examined the effects of repeated ketamine anesthesia during PD 16-20 on tolerance, locomotion, behavioral sensitization, and CPP in adolescent (PD 28-38) and adult (8 week old) mice. The authors also examined the effect of repeated ketamine anesthesia in both male and female mice, as well as gene expression and neuronal excitability in the hippocampus of male mice. Lastly, the authors measured dopamine and DOPAC levels across several brain regions. The authors demonstrate that male mice are susceptible to the effects of repeated ketamine during the late postnatal period as they exhibit enhanced drug reward to low doses of ketamine and subthreshold doses of nicotine. Interestingly, these effects are not evident in female mice. Moreover, they demonstrate that the enhanced effect produced by ketamine is not long-lasting given that 5 weeks after ketamine pretreatment, the ketamine and nicotine enhancement of drug reward is no longer evident. Interestingly, the authors also demonstrate a corresponding change in gene expression in the hippocampus and dysregulated excitability of CA1 hippocampal neurons after repeated ketamine in male mice. Overall, the manuscript is well-written, and the experimental design is solid. This study contributes to our understanding of the effects of early developmental exposure to ketamine on drug reward, of which few studies have been conducted. I also find the inclusion of the adult data (8-week-old mice) to be a strength of the paper. The statistical analysis is appropriate, and the manuscript details enough information to replicate the study. Below I outline my suggestions to improve the clarity of the manuscript.

- The authors' behavioral approaches to examine "addiction behavior" included behavioral sensitization and conditioned place preference (CPP). However, addictive behaviors are often assessed using the self-administration model. CPP, in particular, is a measure of drug reward and not addiction behavior (see Bardo and Bevins, 2000 – cited by the authors). The authors should revise their language to more directly reflect how CPP data is interpreted (i.e., an animal model of drug reward). Similarly, behavioral sensitization has implications for addiction, but it is not an animal model of addictive behaviors. This can be discussed as a limitation of the study. The authors should also consider integrating a discussion of future directions with the self-administration paradigm after repeated ketamine anesthesia during development to address addictive behaviors more directly.

- The abstract should be revised to provide more detail about the purpose and goals of the study, as well as more details of the experimental design and results.

- The last paragraph of the introduction should reflect the goals of the experiments, including details such as examination of ketamine- and nicotine-induced CPP in adolescent and adult male and female mice, behavioral sensitization, etc.

- Minor: Remove "prove" on page 4, line 83.

Reviewer #2 (Remarks to the Author):

The manuscript by Cui and colleagues reports interesting findings characterizing the neurobiological effects of repeated (5 consecutive days) anesthesia, ketamine, during early postnatal development. The authors report that mice exposed to ketamine at anesthetic doses show increased preference for a test compartment associated (paired) with ketamine and nicotine. They further report, interestingly, that the increased responsivity to nicotine remains while that for ketamine attenuates with time. They also report changes (enhanced expression) in genes regulating neuronal excitability within the hippocampus. Overall, they show that only males are susceptible to these ketamine effects. This is an interesting study, however there are a couple suggestions/points of clarification needed.

1. Please state the human equivalent age-range when the ketamine injections were given?
2. Justify the 5 consecutive days of ketamine exposure.
3. How long were the mice (male and females) remained anesthetized each day of exposure?
4. The authors need to address and discuss their findings within the context of the body of literature indicating that females are much more sensitive to the effects of ketamine (Kabbaj and others).
5. To claim that females are not susceptible to "drug memory" the authors would need to demonstrate that the treatment did not affect any other type of cognitive impairment given the effects that ketamine may have on memory.
6. It is important to show that causative relationship between the gene(s) of interest and behavior. Authors may be able to use an inhibitor to show some causality.

Reviewer #3 (Remarks to the Author):

The current manuscript investigated the effects of anesthetic ketamine during neurodevelopment in addictive behaviors in adult life. This manuscript is an important addition to the field; however, there are some important points that need to be addressed.

1. The behavioral experiments are somewhat lacking and not performed rigorously. My biggest concern is the absence of CPP in the control mice in figure 3. The authors suggested that this might be due to the low dose of ketamine at that age. There are papers that used lower doses of ketamine in adult mice and were able to induce effects (eg Zanos et al 2015). Considering the no effect in females I am a bit skeptical about the findings in Fig 2. Did the authors try to replicate their CPP findings from Fig 2?
 2. I am a bit confused on the comparisons for the behavioral sensitization experiment. Are the trials the different days of administration? To see if they were able to induce sensitization as expected they need to compare their findings within group showing that there was an increase in locomotion on the challenge day compare with the previous days. I don't think the authors did that. Of course, it is expected to see an increase in locomotion as they show in Fig 1d, the question is the increase bigger than the previous ketamine administration days? I am not sure what panel E shows. If trials are the different days of administration, then the sensitization paradigm did not work.
 3. The authors refer to sex differences; however, the experiments were not performed together. I would say there are these effects in males but not females but without direct comparison, the authors cannot use the term sex differences. Since studies were not performed together and consequently performing direct comparisons is not correct, I suggest rephrasing the text.
 4. I would have expected the authors to discuss a bit more the differences between males and females. Why is there such a striking difference, especially when there are several papers showing that females have stronger antidepressant effects to ketamine compare with males? Is this the differences in metabolism or the differences in hormones? I suggest discussing these differences a bit more and extrapolate on the possible reasons for such difference.
 5. Also, the authors include the number of cells in the ephys results. This is wrong, the average of each animal should be used. Considering the low n of animals, technically authors cannot run statistics. As is, my confidence in these data is not very high.
- Minor point: The introduction is very vague. The authors referring to differences but not specifically say what the differences are. They need to send a message why this is important but including these vague statements it makes the whole premise of the manuscript weak.

Reviewers' comments:

Reviewer #1 (Remarks to the Author):

This manuscript examined the effects of repeated ketamine anesthesia during PD 16-20 on tolerance, locomotion, behavioral sensitization, and CPP in adolescent (PD 28-38) and adult (8 week old) mice. The authors also examined the effect of repeated ketamine anesthesia in both male and female mice, as well as gene expression and neuronal excitability in the hippocampus of male mice. Lastly, the authors measured dopamine and DOPAC levels across several brain regions. The authors demonstrate that male mice are susceptible to the effects of repeated ketamine during the late postnatal period as they exhibit enhanced drug reward to low doses of ketamine and subthreshold doses of nicotine. Interestingly, these effects are not evident in female mice. Moreover, they demonstrate that the enhanced effect produced by ketamine is not long-lasting given that 5 weeks after ketamine pretreatment, the ketamine and nicotine enhancement of drug reward is no longer evident. Interestingly, the authors also demonstrate a corresponding change in gene expression in the hippocampus and dysregulated excitability of CA1 hippocampal neurons after repeated ketamine in male mice. Overall, the manuscript is well-written, and the experimental design is solid. This study contributes to our understanding of the effects of early developmental exposure to ketamine on drug reward, of which few studies have been conducted. I also find the inclusion of the adult data (8-week-old mice) to be a strength of the paper. The statistical analysis is appropriate, and the manuscript details enough information to replicate the study. Below I outline my suggestions to improve the clarity of the manuscript.

Response:

We thank the reviewer for the summary and constructive comments/suggestions.

1. The authors' behavioral approaches to examine "addiction behavior" included behavioral sensitization and conditioned place preference (CPP). However, addictive behaviors are often assessed using the self-administration model. CPP, in particular, is a measure of drug reward and not addiction behavior (see Bardo and Bevins, 2000 – cited by the authors). The authors should revise their language to more directly reflect how CPP data is interpreted (i.e., an animal model of drug reward). Similarly, behavioral sensitization has implications for addiction, but it is not an animal model of addictive behaviors. This can be discussed as a limitation of the study. The authors should also consider integrating a discussion of future directions with the self-administration paradigm after repeated ketamine anesthesia during development to

address addictive behaviors more directly.

Response:

We thank the reviewer for correcting our interpretation of the CPP results. We have revised our manuscript to properly reflect our CPP data (an animal model of drug reward). We also discussed the need for additional studies using the self-administration paradigm to investigate the association between early ketamine exposures and addiction behavior later in life (page 14, 1st paragraph).

“Lastly, although we discovered significant changes in place preference, the CPP test is not a direct animal model for addiction behavior. The CPP test is an animal model of drug reward, and the drug is passively administered.¹ In contrast, during the self-administration test, the subject must learn a task that produces acute effects from self-drug administration. Thus, the self-administration paradigm is considered a more direct measurement of drug reinforcement and addiction.¹ Future studies using the self-administration paradigm are necessary to address the effects of early repeated ketamine anesthesia regarding addictive behavior.”

2. The abstract should be revised to provide more detail about the purpose and goals of the study, as well as more details of the experimental design and results.

Response:

We thank the reviewer for the comment. We have provided more details in the abstract. Although we also agree that more details are needed, unfortunately, the submission guidelines states that the abstract should be 150 words or fewer.

“Early exposures to anesthetics can cause long-lasting changes in excitatory/inhibitory synaptic transmission (E/I imbalance), an important mechanism for neurodevelopmental disorders. Since E/I imbalance is also involved with addiction, we further investigated possible changes in addiction-related behaviors after multiple ketamine anesthesia in late postnatal mice. Postnatal day (PND) 16 mice received multiple ketamine anesthesia (35 mg·kg⁻¹, 5 days), and behavioral changes were evaluated at PND28 and PND56. Although mice exposed to early anesthesia displayed normal behavioral sensitization, we found significant increases in conditioned place preference to both low-dose ketamine (20 mg·kg⁻¹) and nicotine (0.5 mg·kg⁻¹). By performing transcriptome analysis and whole-cell recordings in the hippocampus, a brain region involved with CPP, we also discovered enhanced neuronal excitability and E/I imbalance in CA1 pyramidal neurons. Interestingly, these changes were not found in female mice. Our results suggest that repeated ketamine anesthesia during neurodevelopment may influence drug reward behavior later in life.”

3. The last paragraph of the introduction should reflect the goals of the experiments, including details such as examination of ketamine- and nicotine-induced CPP in adolescent and adult male and female mice, behavioral sensitization, etc (page 4, last paragraph).

Response:

We thank the reviewer for comment. We have modified the last paragraph of the Introduction to include more details regarding the study (goals and experiments).

“To test our hypothesis, late postnatal mice (postnatal day 16 [PND16]) received multiple anesthetic treatments with ketamine (35 mg·kg⁻¹, i.p.) for five consecutive days. Long-term changes in addiction-related behaviors were evaluated by measuring behavioral sensitization and conditioned place-preference (CPP) to low-dose ketamine (20 mg·kg⁻¹, i.p.) and nicotine (0.5 mg·kg⁻¹ s.c.) in adolescent (PND28) and adult (8 weeks old) mice. Although there was no change in behavioral sensitization, we discovered significant increases in place preference for both ketamine and nicotine in male mice who had received early ketamine anesthesia. Interestingly, such behavioral changes did not occur in female mice. To further identify possible mechanisms underlying the changes in drug reward but not sensitization, we performed transcriptome analysis and whole-cell recordings in the hippocampus, a brain region deeply involved with CPP.^{2, 3, 4} Our results suggest that exposures to ketamine during neurodevelopment can cause long-lasting changes in drug reward, and that sex must be regarded as a significant biological variable when studying the early effects of anesthesia.”

4. Minor: Remove “prove” on page 4, line 83.

Thank you for the correction.

Reviewer #2 (Remarks to the Author):

The manuscript by Cui and colleagues reports interesting findings characterizing the neurobiological effects of repeated (5 consecutive days) anesthesia, ketamine, during early postnatal development. The authors report that mice exposed to ketamine at anesthetic doses show increased preference for a test compartment associated (paired) with ketamine and nicotine. They further report, interestingly, that the increased responsivity to nicotine remains while that for ketamine attenuates with time. They also report changes (enhanced expression) in genes regulating neuronal excitability within the hippocampus. Overall, they show that only males are susceptible to these ketamine effects. This is an interesting study, however there are a couple suggestions/points of clarification needed.

Response:

We thank the reviewer for the excellent summary and constructive comments/suggestions.

1. Please state the human equivalent age-range when the ketamine injections were given?

Response:

We thank the reviewer for the comment and stating the importance of the age of animals.

1) Initial ketamine injections for anesthesia were performed at postnatal day 16 to 20. Based on a model that integrates important neural events to translate neurodevelopment time across mammalian species (www.translatingtime.org), this age is comparable from a neurodevelopmental standpoint to a 6 - 9 months human infant. This age was selected as most clinical studies also focus on infants or very young children. We described this in the Discussion (page 13, 2nd paragraph).

“Although the majority of preclinical studies have been performed at PND7, it has been suggested that this age is comparable to a third-trimester human fetus from a neurodevelopment standpoint (www.translatingtime.org).^{5,6} This may cause discrepancies between preclinical and clinical studies since the majority of clinical studies focused on infants or very young children. Our study performed ketamine anesthesia at PND16-20, an age that has been suggested to be comparable to a 6 to 9 months-old human infant.”

2) To measure the long-lasting changes in reward behavior, the CPP was performed at postnatal day 28 and 56. Based on the same model (www.translatingtime.org), PND28 and PND56 mice are comparable from a neurodevelopmental standpoint to a 16 month-old and 2 year-old child.^{5,6} Thus, although considered adolescent and adult mice, these ages may not

be sufficient to evaluate the long-term effects of early ketamine exposures. We have added this as a limitation in the Discussion (page 13, last paragraph).

“Thirdly, the interval between early ketamine anesthesia and the CPP tests may not have been sufficient to evaluate long-term changes in drug reward. CPP tests were performed at PND28 and PND56. However, these ages may be comparable to a 16-month infant and a 2-year-old child from a neurodevelopmental standpoint.^{5,6} Thus, although considered adolescent and adult mice, these ages may not be sufficient to evaluate the long-term effects of early ketamine anesthesia.”

2. Justify the 5 consecutive days of ketamine exposure.

Response:

We thank the reviewer for the comment. Although there is no standard for studying the neurotoxic effects of multiple anesthetic exposures, most preclinical studies exposed young animals to 3 episodes of anesthetic exposures. Clinically, it is extremely uncommon for a child to experience more than 3 exposures of general anesthesia during childhood. However, ketamine can be used outside the operating room for more frequently repeated short-term procedures or radiographic examinations. Thus, we increased the number of injections (total 5 injections). We have mentioned this in the Methods (page 15, 2nd paragraph).

“Mice in the Anesthesia group received daily intraperitoneal injections of ketamine (35 mg·kg⁻¹) for 5 consecutive days.⁷ Although the majority of preclinical studies exposed young animals to 3 episodes of anesthesia, ketamine is often used outside the operating room for more frequently repeated short-term procedures or radiographic examinations.^{8, 9} Thus, we increased the number of daily injections to 5 days.”

3. How long were the mice (male and females) remained anesthetized each day of exposure?

Response:

We thank the reviewer for the comment. The anesthetic dose of ketamine was based on our previous study, where we found that higher doses occasionally caused prolonged anesthesia or mortality (> 40 mg·kg⁻¹, i.p).⁷ Anesthesia duration was defined as the time from intraperitoneal injection to the regaining of the righting reflex,¹⁰ and was measured in a subset of male and female mice. We described anesthesia durations in Fig 1A, Suppl Fig 6, and Suppl statistics. Intraperitoneal injection of ketamine induced a relatively short duration of anesthesia (loss of righting reflex [LOR] duration) on day 1 in both male and female mice (male = 230 ±

203.4 s, female = 252.5 ± 153.1 s). However, we also discovered that anesthesia duration rapidly declined after repeated ketamine injections (acute tolerance). In fact, LOR did not occur in most male mice after ketamine injection from the third day. Although it may be reasonable to have increased the dose of ketamine to induce loss of righting reflex (anesthetic state) in these mice, this would have caused significant differences in the total amount of ketamine for each mouse. Thus, after careful consideration, we injected mice with an identical dose of ketamine (35 mg/kg) regardless of the development of acute tolerance. However, we also acknowledge that this resulted in mice receiving several injections of ketamine at a sub-anesthetic concentration. We have added this in the Results and Discussion (page 5, 1st paragraph & page 14, 1st paragraph).

“Although ketamine induced a short duration of anesthesia on day 1 (230 ± 203.4 seconds), the duration of anesthesia significantly declined with following injections (Fig. 1a). In fact, due to the development of acute tolerance, LOR did not develop in most mice after day 3 (Fig. 1a).”

“Fourthly, the use of a fixed ketamine dose (35 mg·kg⁻¹) for repetitive anesthesia despite the development of acute tolerance may also act as a limitation. As the purpose of the study was to evaluate the long-term consequences of early ketamine anesthesia, it may have been reasonable to increase the dose of ketamine when LOR did not occur. However, this could complicate data analysis, since each mouse would have received different doses of ketamine”

4. The authors need to address and discuss their findings within the context of the body of literature indicating that females are much more sensitive to the effects of ketamine (Kabbaj and others).

Response:

We thank the reviewer for the comment and highlighting Dr. Kabbai’s research. As stated by the reviewer, previous studies have reported sex-differences in the sensitivity to the effects of ketamine. We agree that this is of significantly importance, and have addressed previous studies with our results in more detail (page 11, last paragraph).

“Although sex is recognized as an important biological factor,¹¹ we did not initially perform experiments in female mice. This was based on our recent study, where we reported non-sex-dependent changes after repeated ketamine anesthesia (two-way ANOVA).⁷ However, another recent study reported conflicting results by showing that repeated ketamine treatment in young mice enhanced place preference to cocaine only in male mice.¹² Interestingly, we also discovered increased drug reward behavior and enhanced hippocampal activity only in male mice. Our results, however, are in contrast with other preclinical studies reporting enhanced

sensitivity to ketamine in female rodents. For instance, studies have reported that the dose-response threshold for ketamine's antidepressant effect was lower in female mice.^{13, 14, 15, 16, 17} One study also reported that the level of sensitization to low-dose ketamine (5 mg·kg⁻¹) was significantly higher in female rats.¹⁴ However, the same study also reported that the difference in sensitization no longer existed when applying a higher dose of ketamine (10 mg·kg⁻¹), and that ketamine-induced CPP occurred only in male rats.¹⁴ Other studies also using ketamine (10 mg·kg⁻¹) have reported longer antidepressant effects in male mice,¹⁶ and that prolonged injections induced antidepressant effects only in male mice.¹⁸ Based on these results, sex differences in ketamine sensitivity may be dose-dependent. Another possibility is that sex differences depend on the behavioral assay. A previous study reported that ketamine 5 mg·kg⁻¹, a dose that induced antidepressant effects and behavioral sensitization in female mice, induced conditioned place aversion rather than inducing CPP.¹³ Another study showed that while ketamine 10 mg·kg⁻¹ induced sensitization in both male and female mice, CPP only developed in male mice.¹⁴ Since distinct brain regions are involved with the diverse effects of ketamine (antidepressant, behavioral sensitization, and drug reward [CPP]), the sex-dependent changes in multiple brain regions after ketamine injection(s) may be involved.^{13, 14, 15, 16, 17, 18}

5. To claim that females are not susceptible to “drug memory” the authors would need to demonstrate that the treatment did not affect any other type of cognitive impairment given the effects that ketamine may have on memory.

Response:

We thank the reviewer for the important comment. By performing the Barnes maze and fear chamber test in female mice, we further showed that contextual learning and memory were not affected by early ketamine anesthesia. Together, our results suggest that female mice are resistant to the increase in drug reward behavior induced by early ketamine anesthesia. We have added this in the Results (page 9, 2nd paragraph).

“While our results suggest that early ketamine anesthesia does not enhance drug reward behavior in female mice, it is important to recognize that CPP measures drug reward by evaluating the association of a specific context and drug stimulus. Thus, to imply that female mice are resistant to the increase in drug reward induced by early ketamine anesthesia, it is necessary to confirm that non-drug-associated memory is also not affected. By performing the Barnes maze and fear chamber test, we further confirmed that non-drug-associated learning and memory were not affected in female mice (Supplementary Fig. 8).”

6. It is important to show that causative relationship between the gene(s) of interest and

behavior. Authors may be able to use an inhibitor to show some causality.

Response:

We thank the reviewer for the comment and agree that the lack of experiments showing a direct causative relationship acts as a significant limitation of the study. Although our transcriptome analysis and electrophysiology data suggest that increased hippocampal excitability may be involved, additional studies specifically inhibiting hippocampal activity are necessary to confirm the causative relationship. While this may be possible through direct drug treatment after surgical implantation of cannulas¹⁹ or optogenetic/chemogenetic inhibition,^{20, 21} we were unable to perform such studies due to lack of expertise and equipment. We have added this as a significant limitation in the Discussion (page 13, 3rd paragraph).

“The first limitation is the lack of evidence confirming a direct causative relationship between the increased hippocampal excitability and changes in reward behavior. Although this can be achieved using direct drug treatment through surgically implanted cannulas¹⁹ or optogenetic/chemogenetic inhibition,^{20, 21} we were unable to perform such studies due to lack of expertise and equipment.”

Reviewer #3 (Remarks to the Author):

The current manuscript investigated the effects of anesthetic ketamine during neurodevelopment in addictive behaviors in adult life. This manuscript is an important addition to the field; however, there are some important points that need to be addressed.

Response:

We thank the reviewer for the constructive comments/suggestions.

1. The behavioral experiments are somewhat lacking and not performed rigorously. My biggest concern is the absence of CPP in the control mice in figure 3. The authors suggested that this might be due to the low dose of ketamine at that age. There are papers that used lower doses of ketamine in adult mice and were able to induce effects (eg Zanos et al 2015). Considering the no effect in females I am a bit skeptical about the findings in Fig 2. Did the authors try to replicate their CPP findings from Fig 2?

Response:

We thank the reviewer for the comment and highlighting Dr. Zanos's research. However, previous studies have also reported inconsistent results regarding ketamine-induced CPP in adult rodents. For instance, several studies have shown that CPP does not develop when using relatively low doses of ketamine ($\leq 5\text{mg/kg}$).^{13, 14} Another study also reported that CPP did not develop at ketamine doses between 0 – 15 mg/kg.²² Based on these reports, it is possible that ketamine-induced CPP may not be as robust as other addictive drugs such as cocaine. Also, unlike most studies that used various visual and tactile cues to distinguish chambers, we used a more simplified CPP apparatus. Such simplicity may have suppressed the development of place preference in the present study. We have added this in the Results (page 7, 1st paragraph & page 12, 1st paragraph).

“Another possibility is that ketamine-induced CPP is not as robust as other addictive drugs, since previous studies have also reported negative results.^{13, 22} The simplicity of our CPP apparatus may also be involved. While most CPP tests use various visual and tactile cues to distinguish chambers, we used a more simplified version (black and white chambers).”

“Our results, however, are in contrast with other preclinical studies reporting enhanced sensitivity to ketamine in female rodents. For instance, studies have reported that the dose-response threshold for ketamine's antidepressant effect was lower in female mice.^{13, 14, 15, 16, 17} One study also reported that the level of sensitization to low-dose ketamine ($5\text{ mg}\cdot\text{kg}^{-1}$) was significantly higher in female rats.¹⁴ However, the same study also reported that the difference

in sensitization no longer existed when applying a higher dose of ketamine (10 mg·kg⁻¹), and that ketamine-induced CPP occurred only in male rats.¹⁴ Other studies also using ketamine (10 mg·kg⁻¹) have reported longer antidepressant effects in male mice,¹⁶ and that prolonged injections induced antidepressant effects only in male mice.¹⁸ Based on these results, sex differences in ketamine sensitivity may be dose-dependent. Another possibility is that sex differences depend on the behavioral assay. A previous study reported that ketamine 5 mg·kg⁻¹, a dose that induced antidepressant effects and behavioral sensitization in female mice, induced conditioned place aversion rather than inducing CPP.¹³ Another study showed that while ketamine 10 mg·kg⁻¹ induced sensitization in both male and female mice, CPP only developed in male mice.¹⁴ Since distinct brain regions are involved with the diverse effects of ketamine (antidepressant, behavioral sensitization, and drug reward [CPP]), the sex-dependent changes in multiple brain regions after ketamine injection(s) may be involved.^{13, 14, 15, 16, 17, 18}

We would also like to mention that all behavioral experiments were rigorously performed in a sound attenuated room by a blind experimenter (Dr. Cui) under the direct supervision of the corresponding author (Dr. Chung), who has considerable experience in behavioral experiments.^{23, 24, 25, 26, 27} However, we also understand the reviewer's concern and agree that replicating our CPP findings in Fig 2 would significantly enhance the credibility of our data. Thus, we repeated the ketamine-induced CPP test at PND28 and was able to replicate our findings. We have added the results in Supplementary Figure 1 (page 6, 2nd paragraph).

2. I am a bit confused on the comparisons for the behavioral sensitization experiment. Are the trials the different days of administration? To see if they were able to induce sensitization as expected they need to compare their findings within group showing that there was an increase in locomotion on the challenge day compare with the previous days. I don't think the authors did that. Of course, it is expected to see an increase in locomotion as they show in Fig 1d, the question is the increase bigger than the previous ketamine administration days? I am not sure what panel E shows. If trials are the different days of administration, then the sensitization paradigm did not work.

Response:

We thank the reviewer for the comment and apologize for the confusion. Behavioral sensitization was induced by injecting ketamine (20 mg/kg) five times at a 4-day interval (5 trials). To provide a more concise description of our results, we modified Figure 1 (page 6, 1st paragraph). Fig 1e shows that early multiple ketamine anesthesia did not affect baseline

activity (Control-Saline vs Anesthesia-Saline). Fig 1f, g shows that the level of sensitization (increases of activity after identical ketamine injections) was comparable between the Control and Anesthesia group (page 5, last paragraph).

“After confirming that the baseline activity was not affected by multiple ketamine anesthesia (Control-Saline group vs. Anesthesia-Saline group, Fig. 1e), mice were repeatedly injected with ketamine in a 4-day interval (Fig. 1d). Behavioral sensitization occurred in both groups, as total activity significantly increased after five identical ketamine injections (Fig. 1f). We also found that the level of sensitization was comparable between the two groups (Fig. 1f, g).”

We would also like to inform the reviewers that we have modified the y axis of Fig 1e-g due to errors during our initial submission ('0' was added, (ex) 10 → 100).

3. The authors refer to sex differences; however, the experiments were not performed together. I would say there are these effects in males but not females but without direct comparison, the authors cannot use the term sex differences. Since studies were not performed together and consequently performing direct comparisons is not correct, I suggest rephrasing the text.

Response:

We thank the reviewer for the correction. We have modified (rephrased) the text and avoided using the term 'sex-dependent' differences, since the experiments were not performed together.

4. I would have expected the authors to discuss a bit more the differences between males and females. Why is there such a striking difference, especially when there are several papers showing that females have stronger antidepressant effects to ketamine compare with males? Is this the differences in metabolism or the differences in hormones? I suggest discussing these differences a bit more and extrapolate on the possible reasons for such difference.

Response:

We thank the reviewer for the comment. As stated by the reviewer, previous studies have reported sex-differences regarding ketamine which should have been discussed in detail. Thus, we have addressed previous studies and our results in the Discussion (page 11, last paragraph).

“Although sex is recognized as an important biological factor,¹¹ we did not initially perform experiments in female mice. This was based on our recent study, where we reported non-sex-

dependent changes after repeated ketamine anesthesia (two-way ANOVA).⁷ However, another recent study reported conflicting results by showing that repeated ketamine treatment in young mice enhanced place preference to cocaine only in male mice.¹² Interestingly, we also discovered increased drug reward behavior and enhanced hippocampal activity only in male mice. Our results, however, are in contrast with other preclinical studies reporting enhanced sensitivity to ketamine in female rodents. For instance, studies have reported that the dose-response threshold for ketamine's antidepressant effect was lower in female mice.^{13, 14, 15, 16, 17} One study also reported that the level of sensitization to low-dose ketamine (5 mg·kg⁻¹) was significantly higher in female rats.¹⁴ However, the same study also reported that the difference in sensitization no longer existed when applying a higher dose of ketamine (10 mg·kg⁻¹), and that ketamine-induced CPP occurred only in male rats.¹⁴ Other studies also using ketamine (10 mg·kg⁻¹) have reported longer antidepressant effects in male mice,¹⁶ and that prolonged injections induced antidepressant effects only in male mice.¹⁸ Based on these results, sex differences in ketamine sensitivity may be dose-dependent. Another possibility is that sex differences depend on the behavioral assay. A previous study reported that ketamine 5 mg·kg⁻¹, a dose that induced antidepressant effects and behavioral sensitization in female mice, induced conditioned place aversion rather than inducing CPP.¹³ Another study showed that while ketamine 10 mg·kg⁻¹ induced sensitization in both male and female mice, CPP only developed in male mice.¹⁴ Since distinct brain regions are involved with the diverse effects of ketamine (antidepressant, behavioral sensitization, and drug reward [CPP]), the sex-dependent changes in multiple brain regions after ketamine injection(s) may be involved.^{13, 14, 15, 16, 17, 18}

5. Also, the authors include the number of cells in the ephys results. This is wrong, the average of each animal should be used. Considering the low n of animals, technically authors cannot run statistics. As is, my confidence in these data is not very high.

Response:

We thank the reviewer for the comment. The statistical approach we used for our electrophysiology results in the present study was based on our previous experience,^{23, 24, 25, 26} and is also commonly used in the published literature regarding electrophysiological recordings. Thus, we considered the present approach reasonable for the purposes of the current study. However, we also understand the reviewer's concern and performed additional tests in order to perform statistics using the average of each animal (at least 2 cells per animal). We have changed all the figures using the average of each animal (Fig 4,5 & Suppl Fig 9, 10). The results were identical with our previous data which used the number of cells.

6. Minor point: The introduction is very vague. The authors referring to differences but not

specifically say what the differences are. They need to send a message why this is important but including these vague statements it makes the whole premise of the manuscript weak.

Response:

We thank the reviewer for the comment. We have modified the Introduction to include more detailed information and stress the importance of the present study (page 3, 1st 2nd paragraph).

“As millions of children receive anesthesia for surgical or diagnostic procedures every year, anesthesia-induced neurotoxicity in the developing brain has received significant research interest in recent decades.²⁸ Preclinical studies have shown that anesthetics commonly used in clinical settings may act as neurotoxins during neurodevelopment, inducing widespread neuronal cell death or excitatory/inhibitory (E/I) imbalance.^{24, 29, 30, 31} The disruption of E/I balance during the critical neurodevelopmental period is of considerable concern, since E/I imbalance is an important mechanism for neurodevelopmental disorders.^{32, 33, 34} In line with such concerns, studies that reported anesthesia-induced E/I imbalance also found long-term behavioral changes in activity, anxiety, sociability, learning, and memory.^{26, 27, 31, 35, 36}

Based on preclinical studies, clinical studies have attempted to identify long-term negative effects of anesthesia in young children. While recent prospective clinical studies suggested that a short anesthetic exposure does not affect general intelligence,^{37, 38, 39} the same studies also reported increased behavioral problems based on parental reports.⁴⁰ Other studies have reported possible associations between early anesthetic exposures and neurodevelopmental disorders (autism, ADHD).^{41, 42} However, it is important to note that anesthesia-induced neurotoxicity was first identified in animal models without direct clinical evidence, making it difficult to speculate how this might manifest in children. Recent studies have acknowledged this fact and performed a wide range of tests to evaluate diverse cognitive and behavioral aspects. While these studies discovered changes in specific behaviors (motor and social linguistic performance, emotions) and executive functions,^{40, 43} many behavioral aspects remain unstudied. One possible, but less evaluated, disorder is addiction. Previous studies have shown that changes in E/I synaptic transmission are deeply involved with addiction.^{44, 45} A single cocaine exposure has been shown capable of affecting AMPA/NMDA receptor currents,⁴⁶ and morphine exposure and withdrawal have been shown to affect GABA signaling.⁴⁷ Thus, it is possible that the synaptic changes due to early anesthesia exposures may also affect addiction behavior.”

References

1. Bardo MT, Bevins RA. Conditioned place preference: what does it add to our preclinical understanding of drug reward? *Psychopharmacology* **153**, 31-43 (2000).
2. Hitchcock LN, Lattal KM. Involvement of the dorsal hippocampus in expression and extinction of cocaine-induced conditioned place preference. *Hippocampus* **28**, 226-238 (2018).
3. Nam MH, *et al.* Activation of Astrocytic μ -Opioid Receptor Causes Conditioned Place Preference. *Cell reports* **28**, 1154-1166.e1155 (2019).
4. Xia L, Nygard SK, Sobczak GG, Hourguettes NJ, Bruchas MR. Dorsal-CA1 Hippocampal Neuronal Ensembles Encode Nicotine-Reward Contextual Associations. *Cell reports* **19**, 2143-2156 (2017).
5. Workman AD, Charvet CJ, Clancy B, Darlington RB, Finlay BL. Modeling transformations of neurodevelopmental sequences across mammalian species. *The Journal of neuroscience : the official journal of the Society for Neuroscience* **33**, 7368-7383 (2013).
6. Charvet CJ. Closing the gap from transcription to the structural connectome enhances the study of connections in the human brain. *Dev Dyn* **249**, 1047-1061 (2020).
7. Lee Y, *et al.* Interval-dependent neurotoxicity after multiple ketamine injections in late postnatal mice. *J Anesth* **35**, 93-101 (2021).
8. Byer DE, Gould AB, Jr. Development of tolerance to ketamine in an infant undergoing repeated anesthesia. *Anesthesiology* **54**, 255-256 (1981).
9. Yalçın Çok O, Evren Eker H, Arıboğan A. Ketamine dosing for sedation during repeated radiotherapy sessions in children. *Turk J Med Sci* **48**, 851-855 (2018).
10. Offenhauser N, *et al.* Increased ethanol resistance and consumption in Eps8 knockout mice correlates with altered actin dynamics. *Cell* **127**, 213-226 (2006).

11. Cabrera OH, Gulvezan T, Symmes B, Quillinan N, Jevtovic-Todorovic V. Sex differences in neurodevelopmental abnormalities caused by early-life anaesthesia exposure: a narrative review. *British journal of anaesthesia* **124**, e81-e91 (2020).
12. Garcia-Carachure I, *et al.* Enduring effects of adolescent ketamine exposure on cocaine- and sucrose-induced reward in male and female C57BL/6 mice. *Neuropsychopharmacology* **45**, 1536-1544 (2020).
13. Strong CE, Schoepfer KJ, Dossat AM, Saland SK, Wright KN, Kabbaj M. Locomotor sensitization to intermittent ketamine administration is associated with nucleus accumbens plasticity in male and female rats. *Neuropharmacology* **121**, 195-203 (2017).
14. Schoepfer KJ, Strong CE, Saland SK, Wright KN, Kabbaj M. Sex- and dose-dependent abuse liability of repeated subanesthetic ketamine in rats. *Physiol Behav* **203**, 60-69 (2019).
15. Carrier N, Kabbaj M. Sex differences in the antidepressant-like effects of ketamine. *Neuropharmacology* **70**, 27-34 (2013).
16. Franceschelli A, Sens J, Herchick S, Thelen C, Pitychoutis PM. Sex differences in the rapid and the sustained antidepressant-like effects of ketamine in stress-naïve and "depressed" mice exposed to chronic mild stress. *Neuroscience* **290**, 49-60 (2015).
17. Dossat AM, Wright KN, Strong CE, Kabbaj M. Behavioral and biochemical sensitivity to low doses of ketamine: Influence of estrous cycle in C57BL/6 mice. *Neuropharmacology* **130**, 30-41 (2018).
18. Thelen C, Sens J, Mauch J, Pandit R, Pitychoutis PM. Repeated ketamine treatment induces sex-specific behavioral and neurochemical effects in mice. *Behav Brain Res* **312**, 305-312 (2016).
19. Otis JM, Fitzgerald MK, Mueller D. Inhibition of hippocampal β -adrenergic receptors impairs retrieval but not reconsolidation of cocaine-associated memory and prevents subsequent reinstatement. *Neuropsychopharmacology* **39**, 303-310 (2014).
20. Sakaguchi M, *et al.* Inhibiting the Activity of CA1 Hippocampal Neurons Prevents the Recall

- of Contextual Fear Memory in Inducible ArchT Transgenic Mice. *PLoS one* **10**, e0130163 (2015).
21. Tuscher JJ, Taxier LR, Fortress AM, Frick KM. Chemogenetic inactivation of the dorsal hippocampus and medial prefrontal cortex, individually and concurrently, impairs object recognition and spatial memory consolidation in female mice. *Neurobiol Learn Mem* **156**, 103-116 (2018).
 22. Parise EM, *et al.* Repeated ketamine exposure induces an enduring resilient phenotype in adolescent and adult rats. *Biological psychiatry* **74**, 750-759 (2013).
 23. Chung W, *et al.* Social deficits in IRSp53 mutant mice improved by NMDAR and mGluR5 suppression. *Nature neuroscience* **18**, 435-443 (2015).
 24. Chung W, *et al.* Sevoflurane Exposure during the Critical Period Affects Synaptic Transmission and Mitochondrial Respiration but Not Long-term Behavior in Mice. *Anesthesiology* **126**, 288-299 (2017).
 25. Kim MH, *et al.* Enhanced NMDA receptor-mediated synaptic transmission, enhanced long-term potentiation, and impaired learning and memory in mice lacking IRSp53. *The Journal of neuroscience : the official journal of the Society for Neuroscience* **29**, 1586-1595 (2009).
 26. Ju X, *et al.* Increasing the interval between repeated anesthetic exposures reduces long-lasting synaptic changes in late postnatal mice. *Journal of neurochemistry*, (2020).
 27. Cui J, *et al.* General Anesthesia During Neurodevelopment Reduces Autistic Behavior in Adult BTBR Mice, a Murine Model of Autism. *Frontiers in cellular neuroscience* **15**, 772047 (2021).
 28. Andropoulos DB, Greene MF. Anesthesia and Developing Brains - Implications of the FDA Warning. *The New England journal of medicine* **376**, 905-907 (2017).
 29. Cabrera OH, Useinovic N, Jevtovic-Todorovic V. Neonatal anesthesia and dysregulation of the epigenome. *Biol Reprod* **105**, 720-734 (2021).

30. Zhou H, Xie Z, Brambrink AM, Yang G. Behavioural impairments after exposure of neonatal mice to propofol are accompanied by reductions in neuronal activity in cortical circuitry. *British journal of anaesthesia* **126**, 1141-1156 (2021).
31. Yang Y, *et al.* Testosterone attenuates sevoflurane-induced tau phosphorylation and cognitive impairment in neonatal male mice. *British journal of anaesthesia* **127**, 929-941 (2021).
32. Del Pino I, Rico B, Marin O. Neural circuit dysfunction in mouse models of neurodevelopmental disorders. *Curr Opin Neurobiol* **48**, 174-182 (2018).
33. Meredith RM. Sensitive and critical periods during neurotypical and aberrant neurodevelopment: a framework for neurodevelopmental disorders. *Neuroscience and biobehavioral reviews* **50**, 180-188 (2015).
34. Lee E, Lee J, Kim E. Excitation/Inhibition Imbalance in Animal Models of Autism Spectrum Disorders. *Biological psychiatry* **81**, 838-847 (2017).
35. Zhao T, *et al.* Prenatal sevoflurane exposure causes neuronal excitatory/inhibitory imbalance in the prefrontal cortex and neurofunctional abnormality in rats. *Neurobiology of disease* **146**, 105121 (2020).
36. Xie L, *et al.* Neonatal sevoflurane exposure induces impulsive behavioral deficit through disrupting excitatory neurons in the medial prefrontal cortex in mice. *Translational psychiatry* **10**, 202 (2020).
37. McCann ME, *et al.* Neurodevelopmental outcome at 5 years of age after general anaesthesia or awake-regional anaesthesia in infancy (GAS): an international, multicentre, randomised, controlled equivalence trial. *Lancet (London, England)* **393**, 664-677 (2019).
38. Sun LS, *et al.* Association Between a Single General Anesthesia Exposure Before Age 36 Months and Neurocognitive Outcomes in Later Childhood. *Jama* **315**, 2312-2320 (2016).
39. Warner DO, *et al.* Neuropsychological and Behavioral Outcomes after Exposure of Young Children to Procedures Requiring General Anesthesia: The Mayo Anesthesia Safety in Kids

(MASK) Study. *Anesthesiology* **129**, 89-105 (2018).

40. Ing C, *et al.* Prospectively assessed neurodevelopmental outcomes in studies of anaesthetic neurotoxicity in children: a systematic review and meta-analysis. *British journal of anaesthesia* **126**, 433-444 (2021).
41. Shi Y, *et al.* Moderators of the association between attention-deficit/hyperactivity disorder and exposure to anaesthesia and surgery in children. *British journal of anaesthesia* **127**, 722-728 (2021).
42. DiMaggio C, Sun LS, Li G. Early childhood exposure to anesthesia and risk of developmental and behavioral disorders in a sibling birth cohort. *Anesthesia and analgesia* **113**, 1143-1151 (2011).
43. Walkden GJ, Gill H, Davies NM, Peters AE, Wright I, Pickering AE. Early Childhood General Anesthesia and Neurodevelopmental Outcomes in the Avon Longitudinal Study of Parents and Children Birth Cohort. *Anesthesiology* **133**, 1007-1020 (2020).
44. Lüscher C, Malenka RC. Drug-evoked synaptic plasticity in addiction: from molecular changes to circuit remodeling. *Neuron* **69**, 650-663 (2011).
45. van Huijstee AN, Mansvelder HD. Glutamatergic synaptic plasticity in the mesocorticolimbic system in addiction. *Frontiers in cellular neuroscience* **8**, 466 (2014).
46. Ungless MA, Whistler JL, Malenka RC, Bonci A. Single cocaine exposure in vivo induces long-term potentiation in dopamine neurons. *Nature* **411**, 583-587 (2001).
47. Bajo M, Madamba SG, Roberto M, Siggins GR. Acute morphine alters GABAergic transmission in the central amygdala during naloxone-precipitated morphine withdrawal: role of cyclic AMP. *Frontiers in Integrative Neuroscience* **8**, (2014).

REVIEWERS' COMMENTS:

Reviewer #2 (Remarks to the Author):

The authors have addressed my comments/concerns to satisfaction. The manuscript reads much better and the authors have added careful caveats for the reader to be aware of the context from which the results should be interpreted it.

Reviewer #3 (Remarks to the Author):

I thank the authors for their responses. They have addressed the majority of my concerns. However, I want to point out that when the authors cite papers to support their findings, they should cite papers that used similar protocol as they did. I am referring to my first point about the CPP. Reference 13 did conditioning sessions every 3 days and the test was performed 3 days after the last conditioning day so by no means this is comparable to what the authors did. In Ref 14 they actually showed development of CPP using 10 mg/kg which is lower than what the authors used. Those 2 papers are from the same lab (Kabbaj) which also supports my previous point that what they did on ref 13 was way different to compare it with the regular CPP protocol. The only ref that supports the authors findings is ref 22, however, by the look of the graph at 0mg/kg the rats demonstrated aversion which indicates something fundamentally wrong with the control. In the author's manuscript, I suggest to only leave ref 22 to support their findings. I strongly encourage the authors to have a careful read of the manuscript and check whether the citations support their statements. Other than that, I am happy with the rest of the additions and answers.

REVIEWERS' COMMENTS:

Reviewer #2 (Remarks to the Author):

The authors have addressed my comments/concerns to satisfaction. The manuscript reads much better and the authors have added careful caveats for the reader to be aware of the context from which the results should be interpreted it.

We would like to thank the reviewer for providing important comments and suggestions that have immensely improved our manuscript.

Reviewer #3 (Remarks to the Author):

I thank the authors for their responses. They have addressed the majority of my concerns. However, I want to point out that when the authors cite papers to support their findings, they should cite papers that used similar protocol as they did. I am referring to my first point about the CPP. Reference 13 did conditioning sessions every 3 days and the test was performed 3 days after the last conditioning day so by no means this is comparable to what the authors did. In Ref 14 they actually showed development of CPP using 10 mg/kg which is lower than what the authors used. Those 2 papers are from the same lab (Kabbaj) which also supports my previous point that what they did on ref 13 was way different to compare it with the regular CPP protocol. The only ref that supports the authors findings is ref 22, however, by the look of the graph at 0mg/kg the rats demonstrated aversion which indicates something fundamentally wrong with the control. In the author's manuscript, I suggest to only leave ref 22 to support their findings. I strongly encourage the authors to have a carefull read of the manuscript and check whether the citations support their statements. Other than that, I am happy with the rest of the additions and answers.

We thank the reviewer for providing important insights that have immensely improved our manuscript, and also apologize for not carefully checking the references. We very much agree that only references relevant to the present should be used. Thus, as the reviewer has suggested, we have carefully read the manuscript and made several modifications (marked in blue). We would like to thank the reviewer again for looking

at our manuscript in great detail, and hope the modifications are acceptable.